# Towards Optimal Robustness in Learning-Augmented Paging

**Peng Chen** [1]   **Hailiang Zhao** [1]   **Xueyan Tang** [2]   **Yixuan Wang** [3]   **Shuiguang Deng** [1]

## Abstract

Learning-augmented paging has been extensively studied in recent years. A key advantage over naive ML-based approaches is *bounded robustness*, which guarantees worst-case performance even when predictions are inaccurate, making these algorithms valuable for real-world systems. Prior work achieves robustness bounds of $2H_k + O(1)$ in the randomized setting, leaving a gap to the optimal competitive ratio $H_k$.

In this paper, we study how to close this gap. We begin by reviewing online optimality and proving a new property of the latest $H_k$-competitive algorithm, which facilitates our analysis in the learning-augmented setting. Then, we review existing learning-augmented paging algorithms and introduce a unifying primitive, the *relative prediction budget*, which captures the essence of establishing robustness and reveals that prior algorithms either overuse or underutilize predictions. Guided by the above analysis, we develop a new framework that achieves the best-possible robustness up to an additive constant for learning-augmented paging: $H_k + O(1)$. Experiments further demonstrate strong practical performance.

## 1. Introduction

The idea of ML for systems has been prevalent in both academic and industrial communities in the past few years, with many learned strategies emerging, such as ML-based paging, scheduling, and resource allocation. However, in practice, the hardest part of applying ML in systems may not be achieving high average accuracy, but coping with the inevitable failures that surface after deployment, especially in production environments. This suggests designing systems that treat predictions as helpful but fallible signals. Motivated by these considerations, an interdisciplinary research direction known as algorithms with predictions (ALPS) or learning-augmented algorithms has flourished following the seminal works of (Lykouris & Vassilvitskii, 2018) and (Kraska et al., 2018). This line of research spans data structures (Kraska et al., 2018; Mitzenmacher, 2018), paging and caching (Lykouris & Vassilvitskii, 2018; Wei, 2020; Rohatgi, 2020; Sadek & Elias, 2024), the Bahncard problem (Drygala et al., 2023; Zhao et al., 2024), ski rental (Purohit et al., 2018; Antoniadis et al., 2021), graph algorithms (Antoniadis et al., 2024; DePavia et al., 2024; He & Li, 2026), and online knapsack (Im et al., 2021; Zeynali et al., 2021; Lechowicz et al., 2024; Daneshvaramoli et al., 2025), among many others.

In this paper, we focus on approaching the optimal robustness bound for learning-augmented paging, a prototypical problem of this area.

**Paging.** Online paging is a canonical online decision problem, arising whenever a system must serve a stream of page requests using a small and fast cache. Formally, we are given a cache of size $k$ and must process page requests online, i.e., without any knowledge of future requests. Requests are processed online: a *hit* costs nothing, while a *miss* loads the page and possibly evicts a resident page, and the total cost is the number of misses.

A classic offline-optimal algorithm (Belady, 1966) shows that *Belady's rule*, evicting the page whose next request occurs latest in the future, minimizes the number of misses. The online setting is fundamentally harder, since the algorithm has no access to future requests. In particular, any deterministic paging algorithm has a competitive ratio at least $k$ (Sleator & Tarjan, 1985), and LRU is $k$-competitive. With randomization, MARKER (Fiat et al., 1991) achieves a competitive ratio of $2H_k - 1$, while achieving the optimal $H_k$-competitive ratio requires a more careful design.[1] PARTITION (McGeoch & Sleator, 1991) was the first algorithm to achieve $H_k$-competitiveness, but it was later characterized as counterintuitive and difficult to interpret (Achlioptas et al., 2000). Subsequent work proposed EQUITABLE (Achlioptas et al., 2000), K_EQUITABLE (Bein et al., 2011), and ON-

[1]Zhejiang University, Hangzhou, China [2]Nanyang Technological University, Singapore [3]Nanjing University of Aeronautics and Astronautics, Nanjing, China. Correspondence to: Hailiang Zhao <hliangzhao@zju.edu.cn>, Shuiguang Deng <dengsg@zju.edu.cn>.

*Proceedings of the $43^{rd}$ International Conference on Machine Learning*, Seoul, South Korea. PMLR 306, 2026. Copyright 2026 by the author(s).

---

[1]$H_k = \sum_{i=1}^{k} \frac{1}{i}$ and $\ln(k+1) \le H_k \le \ln k + 1$.

LINEMIN (Brodal et al., 2015), which match the optimal $H_k$-competitive ratio by tracking the online optimum via a key technique called the *work function*.

**Learning-augmented paging.** Motivated by the success of machine learning on real-world workloads, *learning-augmented algorithms* enrich classical online algorithms with predictions produced by an ML model, while still seeking formal guarantees when predictions are unreliable. The goal is to achieve *near-optimal* performance when predictions are accurate, while retaining a meaningful worst-case bound close to a classical heuristic regardless of prediction quality. A standard vocabulary has emerged around three desiderata: *consistency* (performance under perfect predictions), *robustness* (performance under arbitrarily bad predictions), and *smoothness* (how gracefully performance degrades with prediction error).

In paging, a widely studied type of prediction estimates future reuse, i.e., the next-arrival time of each page, enabling eviction rules that mimic Belady's algorithm when predictions are accurate. The sum of $\ell_1$ distances between predictions and the ground truth is commonly used to measure prediction error, which in turn enables smoothness analyses (Lykouris & Vassilvitskii, 2018; Rohatgi, 2020).

### 1.1. Related Work

The integration of machine learning models to enhance caching performance has been explored in both theory and practice. While prior works have built learning-based caching systems (Vietri et al., 2018), this paper focuses on the theoretical advances in paging algorithms augmented with machine learning.

The first learning-augmented paging algorithm, BLINDORACLE (Lykouris & Vassilvitskii, 2018), evicts the page predicted to be used farthest in the future. It is 1-consistent under perfect predictions, but can be arbitrarily bad when predictions are inaccurate. This brittleness has motivated a broad class of follow-up works. PREDICTIVEMARKER (Lykouris & Vassilvitskii, 2018) incorporates predictions into the MARKER algorithm and achieves $4H_k$-robustness, while LMARKER (Rohatgi, 2020) attains $(2H_k + 4)$-robustness. TRUST&DOUBT (Antoniadis et al., 2023b) uses MARKER-style ideas more carefully to achieve consistency close to (or equal to) 1, whereas many other MARKER-based algorithms sacrifice consistency and remain at least 2-competitive even with perfect predictions. However, MARKER's marking technique inherently limits the best achievable robustness to $2H_k + O(1)$.

Other learning-augmented paging algorithms run a prediction-driven policy and fall back to a robust baseline when errors are detected. BLINDORACLE&LRU (Wei, 2020) yields 2$k$-robustness and 2-consistency. F&R and

F&R (FITF) (Sadek & Elias, 2024) first achieves both 1-consistency and $\mathcal{O}(\log k)$-robustness, but with a non-optimal leading constant factor.

Overall, the gap between existing methods and the online optimal competitive ratio bound $H_k$ motivates us to explore how to approach this optimal bound. Along the way, we distill the unifying principle behind methods for bounded robustness, which in turn reveals a methodology that existing algorithms typically neglect or have not thoroughly studied.

### 1.2. Our Contributions

In this work, we ask two questions: *How can we approach optimal robustness, and at what cost? What is the essence of establishing bounded robustness around the competitive ratio of a classical baseline?*

We answer both the above questions and make the following contributions:

1. We summarize the insights behind online optimality and prove a new property of ONLINEMIN, the latest $H_k$-competitive algorithm. These results broadly guide learning-augmented designs to leverage online optimality to achieve optimal robustness and optimal consistency simultaneously.

2. We then review how existing learning-augmented paging algorithms establish bounded robustness, and introduce the *relative prediction budget*, a primitive that captures their underlying methodologies and reveals that there remains room to better utilize predictions.

3. Building on the above, we propose a new learning-augmented algorithmic framework, RPB-ONOPT. Using this framework, RPB-OM achieves 1-consistency and optimal robustness up to an additive constant, i.e., $H_k + O(1)$. The experiments demonstrate the benefits of our improved robustness guarantees and the stronger designs guided by the relative prediction budget.

## 2. Preliminaries

**Problem setup.** The paging problem consists of a universe of pages $\mathcal{P}$ and a cache that can store at most $k$ pages. An input is a request sequence $\sigma = \langle r_1, ..., r_n \rangle$, revealed online. Each request $r_i$ asks for a page $p_i \in \mathcal{P}$. If the requested $p_i$ is in the cache, $r_i$ incurs a *cache hit*; otherwise, $r_i$ incurs a *cache miss* and the algorithm must load $p_i$ into the cache, evicting a page if the cache is full. The goal is to minimize the total number of cache misses over $\sigma$.

**Metrics.** An *online* algorithm makes eviction decisions without knowledge of future requests, whereas an *offline* algorithm knows $\sigma$ in advance. We refer to a spe-

cific state of the cache as a *cache configuration*. For any algorithm A starting from cache configuration $C$, let $cost(A, C, \sigma)$ denote its cost on $\sigma$, defined as the number of cache misses (and for randomized algorithms, we consider $\mathbb{E}[cost(A, C, \sigma)]$). Note that any algorithm incurs the same cost as OPT when serving $k$ distinct pages starting from an initially empty cache. Conceptually, let this be the warm-up process, after which the cache becomes full for the first time. We omit $C$ when it is the initial empty cache.

Online algorithms are compared to the offline optimal algorithm OPT via the *competitive ratio*. Algorithm A is *$\alpha$-competitive* if there exists a constant $c$ (independent of $\sigma$) such that, for all sequences $\sigma$,

$$cost(A, \sigma) \leq \alpha \cdot cost(OPT, \sigma) + c. \tag{1}$$

In this paper, when the sequence is clear from context, we write $cost(A)$ and $cost(OPT)$, omitting $\sigma$ for brevity.

A learning-augmented caching algorithm receives predictions about future requests, which may be inaccurate. Its guarantees are typically summarized by three components:

1. *consistency $\gamma$*: the competitive ratio under perfect predictions.

2. *robustness $\delta$*: the competitive ratio under adversarial (arbitrarily bad) predictions.

3. *smoothness $f(\eta)$*: a function that characterizes how the competitive ratio depends on the prediction error $\eta$.

## 3. Online Optimality in Paging

To approach the optimal robustness bound, we first comprehensively investigate the structural design underlying online-optimal algorithms, and derive observations that are critical for our new algorithms.

### 3.1. Work Function

Work functions track the current optimal solution and play a central role in online algorithm analysis (Lund & Reingold, 1994; Lund et al., 1994; Chrobak et al., 1997; Irani & Seiden, 1998; Koutsoupias & Papadimitriou, 2000). To distinguish this from the offline optimum, we use "current optimal" for the optimum on the prefix observed so far.

In paging, a work function $\omega$ assigns to each cache configuration the minimum cost needed to serve a request sequence from the beginning and end in that cache configuration. Here, a cache configuration denotes the set of pages stored in the cache. Formally, for $\sigma = \langle r_1, \ldots, r_n \rangle$, let $\omega_\sigma$ be the work function after serving $\sigma$, where $\omega_\sigma(X)$ denotes the minimum cost of serving $\sigma$ and ending in cache configuration $X$. Upon the arrival of a new request $r_{n+1}$ that

requests page $p$, the work function admits the following dynamic-programming update:

$$\omega_{\sigma'}(X) = \min \left\{ \omega_\sigma(X), \ 1 + \min_{x \neq p} \omega_\sigma(X \setminus \{p\} \cup \{x\}) \right\} \tag{2}$$

for any $X$ with $p \in X$, where $\sigma' = \langle r_1, \ldots, r_n, r_{n+1} \rangle$.

When the underlying request sequence $\sigma$ is clear from context, we write $\omega$ instead of $\omega_\sigma$. We use $\omega^r$ and $\omega^\delta$ to denote the work functions obtained from $\omega$ after serving a request $r$ and a request sequence $\delta$, respectively. We denote by $\min(\omega) := \min_X \omega(X)$ the minimum value of the work function over all configurations, which equals the optimal cost of serving the sequence regardless of the final configuration.

**Definition 1.** *A configuration $X$ is called valid iff $\omega(X) - \min(\omega) = 0$.*

**Characteristics.** Koutsoupias & Papadimitriou (2000) show that $\omega$ can be represented by a sequence of layers, which can be updated online. Specifically, layers are represented by $k + 1$ disjoint page sets $(L_0, \ldots, L_k)$. After the warm-up process, each layer $L_i, i > 0$ consists of exactly one page. The following update rules describe how the layers evolve after serving a new request to page $p$:

$$\begin{cases} (L_0 \setminus \{p\}, L_1, \ldots, L_{k-1} \cup L_k, \{p\}) & \text{if } p \in L_0, \\ (L_0, \ldots, L_{i-1} \cup L_i \setminus \{p\}, \ldots, L_k, \{p\}) & \text{if } p \in L_i, i > 0. \end{cases} \tag{3}$$

The *support* of $\omega$ is defined as $S(\omega) = L_1 \cup \ldots \cup L_k$. The layer $L_0$ contains all pages outside the support.

The layer notation provides informative summaries of the optimal solutions for the requests observed so far. In particular, any such solution ends in a valid configuration, and its support satisfies the following lemma from Koutsoupias & Papadimitriou (2000):

**Lemma 3.1.** *A configuration $C$ is valid iff $|C \cap (L_{k-j+1} \cup \cdots \cup L_k)| \geq j$ for all $1 \leq j \leq k$.*

This naturally leads to the following prefix constraints.

**Corollary 3.2.** *A configuration $C$ is valid iff $|C \cap (L_1 \cup \cdots \cup L_j)| \leq j$ for all $1 \leq j \leq k$.*

We refer to a layer containing a single page as a *singleton*. The support then partitions into two sets: (1) the *revealed pages*, $R(\omega) = L_x \cup \cdots \cup L_k$, where $x$ is the smallest index such that all suffix layers $L_x, \ldots, L_k$ are singletons; and (2) the *unrevealed pages*, $N(\omega) = L_1 \cup \cdots \cup L_{x-1}$. We refer to layers containing unrevealed pages as *unrevealed layers*.

There is a *valid configuration construction procedure*: traversing the layers from $L_k$ down to $L_1$ and selecting pages to satisfy Lemma 3.1. The procedure first collects all revealed pages and then adds a subset of unrevealed pages. Based on the above, we obtain the following corollaries:

**Corollary 3.3.** *A valid configuration contains all revealed pages and contains no page in $L_0$.*

**Corollary 3.4.** *By* (3)*, an offline-optimal algorithm transitions between valid configurations after each request. Consequently, it incurs a cache miss if and only if the requested page lies in $L_0$.*

**Corollary 3.5.** *An unrevealed page appears in at least one valid configuration, but it may not be present in the current cache configuration of an offline-optimal algorithm.*

According to the above corollaries, $S(\omega)$ consists of all pages that may appear in some valid configuration; requesting any such page does not increase the current optimal cost, i.e., the value of the work function. A request that does not increase the optimal cost is called a *lazy request* in the literature (Achlioptas et al., 2000), which is actually a request to a page in $S(\omega)$. A lazy request to an unrevealed page is called a *lazy adversary request*. Each lazy adversary request reduces the number of layers containing unrevealed pages by one. Assume that there is an adversary that issues a consecutive request sequence consisting only of lazy adversary requests from now until no unrevealed pages remain in the support. We call such a request sequence a *lazy adversary strategy*.

**Observation 1: Uncertainty in valid configuration transitions.** Due to the dynamic-programming nature of paging, one can track the global optimum by maintaining all local optima. The set of valid configurations, denoted as $\mathcal{V}$, encapsulates all possible configurations that the offline optimum can be in. We note that the size of the valid-configuration set, $|\mathcal{V}|$, captures the *uncertainty* about the offline optimum's current configuration and changes with the type of request.

A request to a page $p \in L_0$ resets the support: it makes all pages in the original $S(\omega)$ unrevealed according to the update rules (3). Consequently, by Lemma 3.1, $|\mathcal{V}|$ increases significantly. Moreover, any page in the original support $S(\omega)$ may be absent from the offline optimum's current configuration. Therefore, such a request introduces a large increase in uncertainty.

On the other hand, for a request to a page $p \in S(\omega)$, if the request is to a page $p \in R(\omega)$, the set of valid configurations remains unchanged. If $p \in N(\omega)$, it becomes revealed. This decreases $|\mathcal{V}|$, thereby narrowing the uncertainty. Note that when all pages in $S(\omega)$ are revealed, the only valid configuration coincides with the offline optimum's configuration.

**Observation 2: Valid configurations and optimality.** The set $\mathcal{V}$ is algorithm-independent and transitions online based only on the requests observed so far, without access to future requests. Since the offline optimum can currently be in any valid configuration in $\mathcal{V}$ depending on future requests, this necessarily creates a gap between the offline and online optima.

This suggests an improvement to reduce this gap by leveraging machine-learned predictions of future requests to narrow the set of valid configurations that the offline optimum may be in, thereby reducing uncertainty and guiding eviction decisions.

### 3.2. Online-Optimal Algorithms

EQUITABLE (Achlioptas et al., 2000) is a randomized online-optimal algorithm. Formally, EQUITABLE maintains a distribution $\Pi_t$ over all valid configurations $\mathcal{V}$ at time $t$, where the probability that the current configuration is $X$ is given by

$$\Pi_t(X) := P_X(\omega_t),$$

where $\omega_t$ denotes the work function at time $t$, and $P_X(\omega_t)$ is defined as the probability that $X$ is the final configuration $X_k$ produced by the following randomized selection process. Initialize $X_0 = \emptyset$. Let $\omega_t^{X_m}$ denote the work function obtained from $\omega_t$ after serving the sequence of requests $x_1, \ldots, x_m$. For $i = 1, ..., k$, sample

$$x_i \sim \text{Uniform}\Big(S(\omega_t^{X_{i-1}}) \setminus X_{i-1}\Big), \; X_i := X_{i-1} \cup \{x_i\}.$$

This construction yields the following lemma, which implies that the adversary cannot benefit from choosing one lazy adversary strategy over another, and it is key to establishing the $H_k$ competitive ratio of EQUITABLE.

**Lemma 3.6.** *(Lemma 4 in Achlioptas et al. (2000)) The expected cost of* EQUITABLE *is the same against all lazy adversary strategies for the current $\omega$.*

The configuration update rule in Achlioptas et al. (2000) can be described as a distribution transition kernel $K(\omega_t, \omega_{t'})$ for serving the next request at time $t'$. It requires simulating the randomized selection process and computing all probabilities, which takes $O(k^2)$ time. A distributional transition example of EQUITABLE is shown in Figure 1.

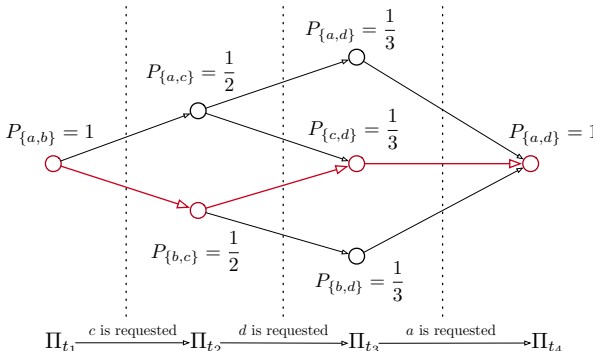

*Figure 1.* An example of EQUITABLE's distributional transitions (black) and ONLINEMIN's update process (red), starting from the configuration $a, b$ and serving requests to $c$, $d$, and $a$ sequentially.

Repeated requests to $L_0$ can make the support grow in EQUITABLE, so *forgiveness* is used to approximate the work

function and cap the size of the support as discussed in detail in Appendix A.

**Fast Implementations.** Brodal et al. (2015) proposed an efficient implementation called ONLINEMIN, which maintains a single configuration via a selection procedure based on random priorities that generates each configuration with the same probability as in EQUITABLE's distribution. We abbreviate ONLINEMIN as OM for the remainder of the paper.

Specifically, OM iteratively constructs sets $C_0, \ldots, C_k$ from the layer partition of $\omega$ by setting $C_0 = \emptyset$ and, for $j = 1, \ldots, k$, $C_j = \max_j(C_{j-1} \cup L_j)$, where $\max_j(S)$ returns the size-$j$ subset of $S$ with the highest priorities. Thus, all revealed pages lie in $C_k$, and $C_k$ is the current configuration of OM. This yields an efficient online update of $C_k$ upon a request to page $p$:

1. If $p \in C_k$: $C'_k = C_k$.

2. If $p \notin C_k$ and $p \in L_0$: $C'_k = C_k \setminus \min(C_k) \cup \{p\}$.

3. If $p \notin C_k$ and $p \in L_i, i > 0$: $C'_k = C_k \setminus \min(C_j) \cup \{p\}$, where $j = \min\{x \geq i \mid |C_x \cap C_k| = x\}$.

This yields a cache update rule with $\mathcal{O}(\log k)$ computation per request. In short, rather than explicitly maintaining a distribution, OM efficiently maintains and updates a single valid configuration sampled from the same distribution as EQUITABLE, as illustrated in Figure 1.

**ONOPT.** Building on OM, Moruz & Negoescu (2012) propose a class of algorithms called ONOPT as shown in Algorithm 1. ONOPT adopts OM's framework while allowing a more flexible priority assignment. OM itself is an instance of ONOPT.

---

**Algorithm 1** ONOPT

---

1: $S(\omega) :=$ the support of the current work function
2: $C :=$ the current cache configuration
3: **while** receiving a request to page $p$ **do**
4:   **if** $p \notin C$ **then**
5:     **if** $p \in L_0$ **then**
6:       Evict a page $x \in C$ with the lowest priority
7:     **else if** $p \in L_i, i > 0$ **then**
8:       Apply OM's rule to select eviction candidates: $E \leftarrow C \cap \bigcup_{l=1}^{z} L_l$, where $z = \min\Big\{ j \in \{i, \ldots, k\} \mid |C \cap \bigcup_{l=1}^{j} L_l| = j \Big\}$
9:       Evict a page $x \in E$ with the lowest self-defined priority
10:   Cache page $p$ and update $C$

---

**Observation 3: Tracking valid configurations online.** At a high level, OM provides an efficient procedure for tracking

valid configurations online. ONOPT builds on this idea to define a broad class of algorithms that always remain in a valid configuration. Building on ONOPT, one can achieve $H_k$-competitiveness.

This naturally motivates incorporating predictions into ONOPT to guide evictions. We find that a new learning-augmented design is needed to fully exploit relatively accurate predictions while adding only an $O(1)$ term to the competitive ratio of a robust baseline, thereby achieving $H_k + O(1)$ robustness. Note that $H_k + O(1)$ is optimal up to an additive constant: to obtain $H_k$ robustness, we would have to follow the original $H_k$-competitive online algorithm exactly, without any use of predictions, which falls outside the learning-augmented setting.

## 4. Relative Prediction Budget

In this section, we review existing learning-augmented paging algorithms and introduce a new notion that captures their core methodologies while also revealing their limitations.

A central difficulty in learning-augmented design is that predictions can be extremely helpful when accurate, yet arbitrarily harmful when adversarial. Robust designs therefore rely on *gating* prediction-driven actions to prevent prediction errors from causing unbounded costs. We make this idea explicit via the *Relative Prediction Budget (RPB)*: fix a robust baseline algorithm $\mathcal{A}$ and maintain a nonnegative budget $B_t$. The algorithm may deviate from $\mathcal{A}$ and take prediction-driven actions only when it can pay from $B_t$. Prior learning-augmented paging algorithms can be viewed as implicitly maintaining and updating such a budget *relative to $\mathcal{A}$*, but with different earning rules.

For example, BLINDORACLE&LRU (Wei, 2020) switches between BLINDORACLE and LRU, always following the one with smaller cost on the observed prefix. Implicitly, it earns one unit of budget *globally* at each step whenever BLINDORACLE's cumulative prefix cost is lower *relative* to LRU's, and it spends one unit of budget by following BLINDORACLE. This budget earning rule is coarse-grained, since it depends only on global past performance. As a result, its total cost is at most twice LRU's cost, yielding $2k$-robustness. F&R (Sadek & Elias, 2024) switches to a robust algorithm whenever it performs worse than BELADY at the current step on the observed request sequence. Thus, it earns prediction budget whenever its current-step cost matches BELADY's, and spends it by continuing to follow the predictions. However, detecting the offline optimum's cost in this way requires excessive recomputation.

In contrast, Marker-based methods such as PREDICTIVE-MARKER (Lykouris & Vassilvitskii, 2018) and LMARKER (Rohatgi, 2020) earn budget *locally*, *relative* to the cost of MARKER. Specifically, budget is earned at the events where

the robust baseline MARKER (Fiat et al., 1991) must pay, namely the arrival of a clean request within the current phase, which initiates an eviction chain. PREDICTIVEMARKER earns $H_k$ units of budget per clean request, and continues to follow predictions along the eviction chain until its length exceeds $H_k$. This aggressive earning rule inflates its robustness to $4H_k$, nearly twice the $(2H_k - 1)$-competitiveness of MARKER. In contrast, LMARKER earns only a single unit of budget per clean request, and follows predictions only once per eviction chain. This conservative rule improves robustness to $2H_k + 4$. However, it can underutilize relatively accurate predictions when prediction-driven evictions are actually better than MARKER's random eviction, which limits its practical performance.

**Observation 4: Limitations of existing budget rules.** The above examples show that existing robust learning-augmented algorithms earn prediction budget relative to a robust baseline's cost, ensuring that performance does not deviate too far. However, there exist two limitations:

1. Lack of fine-grained evaluation of algorithm performance: BLINDORACLE&LRU relies only on global performance comparisons, which may underutilize locally accurate predictions and overuse predictions under adversarial inputs, leading to a robustness bound with an extra multiplicative factor of 2.

2. Lack of a fine-grained correlation between the prediction budget and prediction effectiveness: PREDICTIVEMARKER and LMARKER assess the cost caused by prediction errors via an eviction-chain length threshold, and abandon prediction-driven evictions once the threshold is reached. However, their thresholds, i.e., $H_k$ and 1, are fixed and do not adaptively correlate with the effectiveness of prediction-driven evictions, as reflected by past performance. This can lead to either overusing predictions under the $H_k$ threshold when prediction errors are large, resulting in $4H_k$-robustness, or underutilizing predictions under the 1 threshold when predictions are relatively accurate, resulting in weak practical performance.

Since our goal is to keep robustness within an additive constant of the robust baseline's competitive ratio, we adaptively correlate the prediction budget with both prediction effectiveness and the baseline's performance.

## 5. A Learning-Augmented Design

We first present several key conclusions, building on the principles underlying online-optimal algorithms and the insights stated above. These conclusions then naturally lead to our new algorithm.

### 5.1. Benefiting From Predictions

Observations 1 and 2 motivate incorporating predictions upon a request to a page outside the support, i.e., a page in $L_0$. At such a step, the uncertainty is maximized, making accurate predictions particularly valuable for guiding evictions and reducing the cost incurred by lazy requests that may follow. This also creates an opportunity for a learning-augmented algorithm to behave optimally, as shown next.

In Theorem 5.1 (as proved in Appendix B), "perfect predictions" means that the advice can guide the algorithm by prioritizing for eviction the pages that the offline optimum will evict before their next requests, ahead of the remaining pages. Here, "following the predictions" means the algorithm evicts the page following the priorities given by the advice. For example, if the predictions provide next-arrival times, the algorithm evicts the cached page with the largest predicted next-arrival time.

**Theorem 5.1.** *With perfect predictions, a learning-augmented algorithm* ALG *that, upon a cache miss on a request to a page in $L_0$, evicts following the predictions is optimal, and hence 1-consistent.*

We note that Theorem 5.1 also establishes the minimal prediction-usage requirement for achieving 1-consistency. A detailed discussion of prediction usage, along with related prior work, is deferred to Appendix C.

### 5.2. A New Learning-Augmented Framework

Observation 3, together with Section 4, motivates incorporating predictions into the ONOPT framework while designing a new method to earn prediction budget relative to both prediction quality and the offline optimum's cost.

We propose an algorithmic framework RPB-ONOPT (i.e., ONOPT augmented with relative prediction budget), as shown in Algorithm 2. RPB-ONOPT earns prediction budget via a fine-grained comparison between the effectiveness of the algorithm's evictions and the evictions the baseline would have made in the same situation. By doing so, we can both make full use of good predictions and avoid the high costs that may be caused by large prediction errors.

At each $L_0$-miss, the budget is reset to an $O(1)$ constant $\tau$, which allows for prediction-driven evictions during the subsequent lazy adversary requests.

Then, we show that RPB-OM, which integrates OM into RPB-ONOPT, achieves both 1-consistency and $(H_k + O(1))$-robustness. The formal algorithm description of RPB-OM is provided in Appendix H. Specifically, RPB-OM substitutes the self-defined priority in Line 21 in Algorithm 2 with OM's priority. Moreover, RPB-OM sets $\mathcal{G}_{RPB}(Y, \omega) = (U(\omega) \leq (Y + 2)/e - 2)$ and $\mathcal{U}_{RPB}(\omega) = U(\omega)$, where $U(\omega) = k - |R(\omega)|$ is the number of unre-

**Algorithm 2** RPB-ONOPT

1: $\omega :=$ the current work function
2: $C :=$ the current cache configuration
3: $B :=$ the prediction budget
4: $\tau :=$ the $O(1)$ budget value reset at an $L_0$-miss
5: $Y :=$ the eviction-effectiveness metric
6: $V :=$ the eviction candidate set
7: $\mathcal{G}_{RPB}, \mathcal{U}_{RPB} :=$ the RPB gate and update functions
8: **while** receiving a request to page $p$ **do**
9:    **if** $p \notin C$ **then**
10:      **if** $p \in L_0$ **then**
11:        Evict a page $x \in C$ following the predictions
12:        $B \leftarrow \tau$
13:      **else if** $p \in L_i, i > 0$ **then**
14:        Apply OM's rule to select eviction candidates:
        $V \leftarrow C \cap \bigcup_{l=1}^{z} L_l$, where $z = \min\Big\{ j \in$
        $\{i, .., k\} \mid |C \cap \bigcup_{l=1}^{j} L_l| = j\Big\}$
15:        **if** $\mathcal{G}_{RPB}(Y, \omega)$ **then**
16:          $B \leftarrow B + 1$
17:        **if** $B > 0$ **then**
18:          Evict a page $x \in V$ following the predictions
19:          $B \leftarrow B - 1$
20:        **else**
21:          Evict a page $x \in V$ with the lowest self-defined priority
22:    Cache page $p$, update $C$
23:    $Y \leftarrow \mathcal{U}_{RPB}(\omega)$

vealed layers. The intuition behind the choice of the RPB functions is twofold. First, in general, they correlate the effectiveness of predictions with the prediction budget, and the budget is charged only when the previous evictions perform well relative to the robust baseline. Second, we keep the gate $\mathcal{G}_{RPB}$ concise by using a worst-case lower bound on OM's accumulated expected cost since the last miss; this suffices for robustness close to the $H_k$-competitiveness of OM. A finer alternative is to estimate OM's expected cost at each step directly, which leads to different RPB functions but is still computable without simulating OM's cache. Appendix K presents one such design as the hit-credit variant of RPB-OM.

### 5.3. Optimal Consistency

Theorem 5.2 follows from Theorem 5.1.

**Theorem 5.2.** RPB-ONOPT *is* 1-*consistent.*

**Corollary 5.3.** RPB-OM *is* 1-*consistent.*

### 5.4. Robustness Near Optimal Competitiveness

We first present a fundamental lemma below that serves as the basis for subsequent proofs. It applies to all algorithms

in the ONOPT class, including OM and RPB-OM.

Define the difference $D$ between the caches of two algorithms as the number of pages in $C_1$ but not in $C_2$, namely $D = |C_1 \setminus C_2|$. Denote by $\Delta D$ the change of the cache difference that happens when serving a request. Lemma 5.4 characterizes how the cache difference changes, with its proof given in Appendix D.

**Lemma 5.4.** *Suppose that, when serving a lazy adversary request,* ONOPT *incurs a miss, whereas* OM *incurs a miss with probability* $x$. *If* ONOPT *currently evicts a page according to* OM's *priorities, then the change of expected difference* $D$ *between the cache configurations of the two algorithms satisfies* $\Delta D \leq -1 + x$. *Otherwise,* $\Delta D \leq x$.

The robustness analysis of RPB-OM relies on the competitiveness of OM. Brodal et al. (2015) prove OM's optimality indirectly, by showing that its cache configuration distribution matches that of EQUITABLE, whose $H_k$-competitiveness is established via a potential-based argument. The potential function is defined below.

**Definition 2** (OM's potential function)**.** *Let* $\phi(\omega)$ *denote the expected cost incurred by* OM *when serving a lazy adversary strategy from work function* $\omega$.

However, once predictions are incorporated, RPB-OM's configuration may diverge from OM's, making it difficult to analyze RPB-OM's robustness using the original potential function. We therefore revisit OM in Appendix G, where we derive Corollary 5.5 and prove a new property of OM in Lemma 5.6. These results together facilitate the proof of RPB-OM's robustness.

**Corollary 5.5.** *For any* $\omega$, *when* OM *serves a request to* $p \in L_0$, *the potential change satisfies* $\Delta\phi = \phi(\omega^p) - \phi(\omega) \leq H_k - 1$.

**Lemma 5.6.** *For any* $\omega$, *when* OM *serves a lazy adversary request to* $p \in N(\omega)$, *the potential change satisfies* $\Delta\phi = \phi(\omega^p) - \phi(\omega) \leq -1/(U(\omega)+1)$, *where* $U(\omega) = k - |R(\omega)|$ *is the number of unrevealed layers.*

*Proof.* See details in Appendix G.2. ☐

**A New Potential Function.** Let $\omega$ be the current work function and $C(\omega)$ be the current cache configuration of RPB-OM.

**Definition 3** (RPB-OM's potential function)**.** *Let* $\Phi(\omega)$ *be a potential function for* RPB-OM:

$$\Phi(\omega) = \phi(\omega) + D(\omega) + B(\omega),$$

*where* $\phi(\omega)$ *is defined in Definition 2,* $D(\omega)$ *denotes the expected cache difference between* RPB-OM *and* OM, *and* $B(\omega)$ *denotes the current remaining prediction budget of* RPB-OM.

**Theorem 5.7.** *RPB-OM is* $(H_k + O(1))$-*robust.*

*Proof.* Upon an $L_0$-miss, RPB-OM performs a prediction-driven eviction and OM also incurs a miss, so $\Delta D \leq 1$, and $\Delta \phi \leq H_k - 1$ by Corollary 5.5. Moreover, $\Delta B \leq \tau = O(1)$ since $B$ is reset to $\tau$. Thus, (4) holds in this case.

$$\Delta cost(\text{RPB-OM}) + \Delta \Phi \leq (H_k + O(1)) \cdot \Delta cost(\text{OPT}). \tag{4}$$

Next, we prove that the following inequality always holds when serving a lazy request.

$$\Delta cost(\text{RPB-OM}) + \Delta \Phi \leq 0. \tag{5}$$

If the lazy request is to a revealed page, both RPB-OM and OM hit, and the number of unrevealed layers remains unchanged. Hence, $\Delta \phi = \Delta D = \Delta B = 0$, and (5) holds.

Otherwise, the lazy request is to an unrevealed page, i.e., a lazy adversary request. If RPB-OM hits, then $\Delta \phi \leq 0$, $\Delta D \leq 0$ and $\Delta B = 0$, and (5) holds. Then, we prove (5) through case-by-case discussion when RPB-OM misses.

1. Gate-fail: $U(\omega) > (Y+2)/e - 2$. Two subcases arise according to the current prediction budget $B$.

   (a) $B > 0$: RPB-OM performs a prediction-driven eviction, leading to $\Delta B = -1$. Suppose OM misses with probability $x$, then $\Delta \phi \leq -x$. Moreover, $\Delta D \leq x$ by Lemma 5.4. This gives $\Delta \Phi \leq -1$; thus (5) holds.

   (b) $B = 0$: RPB-OM performs an eviction according to OM's priorities. Suppose OM misses with probability $x$; then $\Delta \phi \leq -x$. Moreover, $\Delta D \leq -1 + x$ by Lemma 5.4. Thus, $\Delta \Phi \leq -1$ and (5) holds.

2. Gate-pass: $U(\omega) \leq (Y+2)/e - 2$. In this case, we additionally consider the previous process during which the number of unrevealed layers decreases from $Y$ at RPB-OM's previous miss to $U(\omega)$. During this process, RPB-OM always hits on lazy requests. Thus, the total difference change $\Delta D' \leq 0$, and the budget remains unchanged, i.e., $\Delta B' = 0$. For $\Delta \phi'$, it depends on the change of the number of unrevealed layers by Lemma 5.6. We denote by $\Delta \phi'$ the total potential change during this process. Thus,

$$\begin{aligned} \Delta \phi' &\leq - \sum_{i=U(\omega)+2}^{Y+1} \frac{1}{i} \\ &\leq - \int_{U(\omega)+2}^{Y+2} \frac{dx}{x} \quad \text{since } \frac{1}{x} \geq \int_x^{x+1} \frac{dx}{x} \\ &= - \ln \frac{Y+2}{U(\omega)+2} \\ &\leq -1. \end{aligned} \tag{6}$$

Therefore, during this process, we have

$$\Delta cost(\text{RPB-OM})' + \Delta \Phi' \leq -1, \tag{7}$$

where $\Delta \Phi' = \Delta \phi' + \Delta D' + \Delta B'$.

Now we serve the current request. The budget earned in the current gate-pass step is immediately consumed by the subsequent prediction-driven eviction, so the net budget change is $\Delta B'' = 0$.

Suppose OM misses with probability $x$, so that the current potential change $\Delta \phi'' \leq -x$, and the current difference change $\Delta D'' \leq x$ by Lemma 5.4. Then, we have $\Delta \Phi'' \leq 0$ and

$$\Delta cost(\text{RPB-OM})'' + \Delta \Phi'' \leq 1, \tag{8}$$

where $\Delta \Phi'' = \Delta \phi'' + \Delta D'' + \Delta B''$.

Combining (8) and (7), we obtain (5).

Finally, summing over all requests that incur $L_0$-misses by (4) and lazy requests by (5), we obtain

$$\begin{aligned} cost(\text{RPB-OM}) + \Phi_{final} - \Phi_{initial} \\ \leq (H_k + O(1)) \cdot cost(\text{OPT}), \end{aligned}$$

where $\Phi_{final} \geq 0$, and $\Phi_{initial} = \phi_{initial} + D_{initial} + B_{initial} = 0$ as the initial cache is empty. Therefore, we have

$$cost(\text{RPB-OM}) \leq (H_k + O(1)) \cdot cost(\text{OPT}).$$

This completes the proof. □

The computational overhead of RPB-OM is mainly introduced by layer partition updating, which is $O(\log k)$ per request, consistent with ONOPT and OM (Brodal et al., 2015).

We also derive RPB-OM's smoothness in Theorem 5.8, proved in Appendix H.1. This bound matches the best-known smoothness guarantee among existing robust randomized algorithms, including LMARKER.

**Theorem 5.8.** *RPB-OM is* $O(1 + \log(\eta_1/cost(\text{OPT})))$-*smooth, where $\eta_1$ denotes the total $\ell_1$ error in the predicted next-arrival times.*

## 6. Experiments

**Benchmark Datasets.** Following recent work (Chłędowski et al., 2021), we use real-world traces from the well-known and commonly used SPEC CPU 2006 benchmark (CRC, 2017) to evaluate empirical performance. We set the cache size to 2MB and the associativity to 16, as is standard in the literature for this benchmark. With 64-byte cache lines,

this yields 2,048 sets of 16 lines each, and cache eviction is performed independently within each set ($k = 16$).

**Prediction Setup.** In the experiments, both synthetic and real predictions are used to evaluate algorithms. Synthetic predictions with different levels of noise have been widely used by prior work (Lykouris & Vassilvitskii, 2018; Chłę-dowski et al., 2021; Sadek & Elias, 2024). Specifically, the noisy predicted next-arrival times of pages are provided by adding log-normal noise to the ground truth.

For real predictors, we adopt POPU following Sadek & Elias (2024). POPU (Antoniadis et al., 2023b) is a frequency-based predictor that assumes a page requested in a fraction $p$ of past accesses will reappear after $1/p$ steps. Additionally, we follow the milestone work (Lykouris & Vassilvitskii, 2018) and use PLECO (Anderson et al., 2014) as a real predictor. PLECO is a probability-based predictor that estimates a page's access likelihood $p$ and predicts its next request arriving after $1/p$ steps.

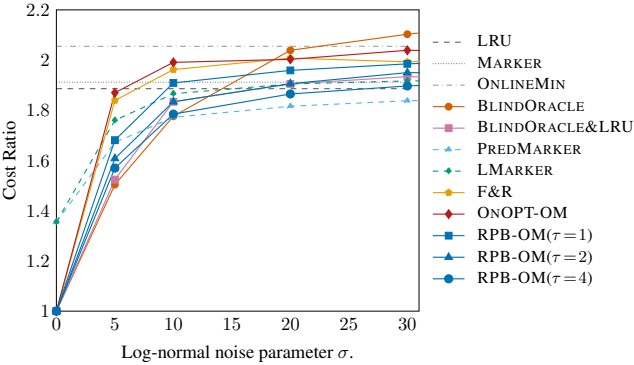

*Figure 2.* Algorithm performance with synthetic predictions of next-arrival times on the bzip dataset.

**Experimental Results.** Figure 2 shows algorithm performance under varying levels of synthetic next-arrival-time prediction noise on the *bzip* dataset from the SPEC CPU 2006 benchmark. Cost ratio denotes the number of misses incurred by an algorithm relative to OPT. Table 1 shows that RPB-OM achieves the best average performance on the SPEC CPU 2006 benchmark under both the PLECO and POPU predictors.

We also compare ONOPT-OM, which incorporates OM into the ONOPT framework without the relative prediction budget design. Specifically, ONOPT-OM uses predictions only on $L_0$-misses and otherwise follows OM's priority within ONOPT. RPB-OM with $\tau = 1$ outperforms ONOPT-OM in both Figure 2 and Table 1. A slightly larger $\tau$ allows RPB-OM to use predictions more aggressively without affecting its asymptotic robustness. BLINDORACLE lacks robustness, while PREDIC-TIVEMARKER and LMARKER lack the ideal 1-consistency, limiting their practical performance when prediction er-

*Table 1.* Average performance of algorithms on the SPEC CPU 2006 benchmark using the PLECO and POPU predictors.

| Algorithm | Cost Ratio ↓ | | Hit Ratio (%) | |
|---|---|---|---|---|
| | Mean | Std | Mean | Std |
| OPT | 1.000 | 0.000 | 34.62 | 25.60 |
| LRU | 1.478 | 0.640 | 14.27 | 20.17 |
| MARKER | 1.392 | 0.361 | 16.51 | 21.13 |
| ONLINEMIN (OM) | 1.396 | 0.329 | 16.30 | 20.12 |
| *With PLECO predictor* | | | | |
| BLINDORACLE | 1.404 | 0.405 | 15.92 | 21.24 |
| F&R | 1.359 | 0.310 | 17.92 | 21.34 |
| LMARKER | 1.335 | 0.260 | 18.11 | 23.18 |
| PREDICTIVEMARKER | 1.335 | 0.261 | 18.10 | 23.20 |
| BLINDORACLE&LRU | 1.286 | 0.239 | 20.53 | 23.55 |
| ONOPT-OM | 1.253 | 0.259 | 22.42 | 23.34 |
| RPB-OM ($\tau = 1$) | **1.249** | 0.268 | **22.75** | 23.29 |
| RPB-OM ($\tau = 2$) | 1.252 | 0.274 | 22.70 | 23.18 |
| RPB-OM ($\tau = 4$) | 1.254 | 0.279 | 22.64 | 23.10 |
| *With POPU predictor* | | | | |
| BLINDORACLE | 1.261 | 0.254 | 21.75 | 23.99 |
| F&R | 1.319 | 0.256 | 19.29 | 22.72 |
| LMARKER | 1.320 | 0.248 | 18.93 | 23.33 |
| PREDICTIVEMARKER | 1.312 | 0.237 | 19.13 | 23.68 |
| BLINDORACLE&LRU | 1.230 | 0.220 | 23.10 | 24.65 |
| ONOPT-OM | 1.237 | 0.234 | 23.76 | 23.00 |
| RPB-OM ($\tau = 1$) | 1.220 | 0.219 | 24.34 | 23.61 |
| RPB-OM ($\tau = 2$) | 1.215 | 0.216 | 24.49 | 23.84 |
| RPB-OM ($\tau = 4$) | **1.213** | 0.215 | **24.55** | 23.93 |

rors are large or small, respectively. RPB-OM exhibits both bounded robustness and 1-consistency across different values of $\tau$. See Appendix I for additional experimental results and Appendix J for the $\tau$-sensitivity analysis. Code is available at https://github.com/Natureal/ICML-Cache-Coliseum.

## 7. Conclusion

In this paper, we provide insights into online optimality that are useful for the learning-augmented setting. Building on these insights, we propose RPB-ONOPT, a framework that enables achieving the best possible robustness for learning-augmented paging algorithms, namely $H_k + O(1)$, thereby closing a long-standing gap in this field. Using this framework, RPB-OM achieves both 1-consistency and $(H_k + O(1))$-robustness. In addition to theoretical results, the experiments validate the strong empirical performance of RPB-OM on real-world traces from the SPEC CPU 2006 benchmark.

Moreover, we introduce the primitive of *relative prediction budget*, which reveals the key mechanism by which existing learning-augmented paging algorithms achieve bounded robustness. Future work may explore how to incorporate relative prediction budget designs into existing algorithms that may currently overuse or underutilize predictions.

## Acknowledgements

We thank the anonymous reviewers for their valuable comments and suggestions. This research was supported in part by the National Key Research and Development Program of China under Grant 2024YFB3309400, the National Science Foundation of China (62502441, 62125206), the Major Program of the National Natural Science Foundation of Zhejiang (LD25F020002), the Singapore Ministry of Education under Academic Research Fund Tier 1 Award RG15/25, and the Zhejiang Key Laboratory Project (2024E10001). Hailiang Zhao's work was supported in part by the Zhejiang University Education Foundation Qizhen Scholar Foundation.

## Impact Statement

This work advances learning-augmented algorithms, which bridge machine learning and theoretical computer science. By improving worst-case paging performance guarantees, it contributes to ML for Systems and may enable more efficient and safe caching systems in practice. We do not foresee immediate negative societal impacts.

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

## A. Forgiveness

Repeated requests to $L_0$ can make the support grow in EQUITABLE, so *forgiveness* is used to approximate the work function and cap the size of the support within $\mathcal{O}(k^2 \log k)$.

K_EQUITABLE (Bein et al., 2011) uses a sharper forgiveness rule and achieves $\mathcal{O}(k)$ space: when $|S(\omega)| = 3k$ and a page $p \in L_0$ is requested, it removes $L_1$ from the support and adds $\{p\}$ as the new $L_k$. It is equivalent to inserting $p$ into $L_1$ and updating $\omega$ as if $p$ had been requested from $L_1$. Specifically, the update rule for K_EQUITABLE, including its forgiveness step, upon a request to page $p$ is as follows:

1. If $p \in L_0$, the update depends on the support size:

$$\begin{cases} (L_0 \backslash \{p\}, L_1, ..., L_{k-1} \cup L_k, \{p\}) & \text{if } |S(\omega)| < 3k, \\ (L_0 \backslash \{p\} \cup L_1, L_2, ..., L_k, \{p\}) & \text{if } |S(\omega)| = 3k. \end{cases}$$

2. If $p \in L_i, i > 0$, the update is the same as EQUITABLE.

The intuition is that a large support indicates that the adversary has already deviated from the worst-case behavior, creating slack between the realized ratio and the worst-case bound of $H_k$. Since future requests cannot recover this slack, the online algorithm can safely deviate from the exact layer update rules while still preserving $H_k$-competitiveness.

## B. Proof of Theorem 5.1

Theorem 5.1. *With perfect predictions, a learning-augmented algorithm* ALG *that, upon a cache miss on a request to a page in $L_0$, evicts following the predictions is optimal, and hence 1-consistent.*

We first show that when predictions are perfect, ALG incurs no cache misses on requests to pages in the support.

Suppose that at time $t$, ALG incurs a cache miss on a request to a page $p$ in the support, and that this is ALG's first miss on requests to pages in the support. By Corollary 3.4, the offline optimum incurs a cache hit at time $t$, implying the page $p$ has been requested before $t$. Therefore, ALG must have incurred a cache miss and evicted $p$ at some time $t' < t$. By assumption, the request that arrived at time $t'$ must have requested a page in $L_0$, and ALG evicted the page $p$ following predictions.

Note that the offline optimum keeps page $p$ in the cache from its previous request that happened before $t'$ until time $t$. Since the predictions are perfect, the only possible reason why ALG evicts $p$ at time $t'$ is that no cached page has a higher eviction priority than $p$. This means that every page cached by ALG at time $t'$ is retained by the offline optimum until its next request, and therefore appears in the offline optimum's current configuration. However, the offline optimum also incurs a cache miss at time $t'$ because the request is to a page in $L_0$, which yields a contradiction.

Therefore, with perfect predictions, ALG incurs cache misses only on requests to pages in $L_0$. By Corollary 3.4, ALG is no worse than the offline optimum and thus is optimal, completing the proof.

## C. Reducing the Use of Predictions

Prior work has investigated how to reduce prediction usage. Antoniadis et al. (2023a) reduces the prediction range of pages when performing eviction. F&R (Sadek & Elias, 2024) recomputes the optimal cost by simulating BELADY on the requests observed so far, and queries the predictor only when both the algorithm and BELADY incur a miss; otherwise, it synchronizes its configuration to BELADY's. Although F&R has high time complexity, it limits the number of calls to the predictor and achieves 1-consistency.

Moreover, 1-consistency has received increasing attention and has been shown to yield superior empirical performance when prediction errors are small (Lykouris & Vassilvitskii, 2018; Sadek & Elias, 2024). The theorem below establishes the minimum prediction-usage requirement for achieving 1-consistency.

Theorem C.1. *To be 1-consistent, a learning-augmented algorithm* ALG *must at least leverage predictions on each request to a page in $L_0$.*

*Proof.* Suppose a request to a page in $L_0$ arrives at time $t$ and ALG incurs a cache miss, but makes its eviction decision without using the predictions. Let $p$ be the page evicted by ALG at time $t$. The adversary can then request $p$ again next,

forcing ALG to incur an additional cache miss. In contrast, the offline optimum evicts some page $q \neq p$ at time $t$, since $p$ has the earliest next-arrival time. Hence, even if ALG is optimal up to time $t$, it incurs strictly more misses than the offline optimum after serving the request to $p$ again. This violates 1-consistency, completing the proof. $\qquad\square$

# D. Proof of Lemma 5.4

**Lemma 5.4.** *Suppose that, when serving a lazy adversary request,* ONOPT *incurs a miss, whereas* OM *incurs a miss with probability* $x$. *If* ONOPT *currently evicts a page according to* OM*'s priorities, then the change of expected difference* $D$ *between the cache configurations of the two algorithms satisfies* $\Delta D \leq -1 + x$. *Otherwise,* $\Delta D \leq x$.

*Proof.* The lemma follows by combining Lemma D.1 and Lemma D.2, which distinguish whether OM misses on the current request. Their proofs are given in Appendix E and Appendix F, respectively.

**Lemma D.1.** *Suppose that when serving a lazy adversary request,* ONOPT *misses, whereas* OM *hits. If* ONOPT *currently evicts a page according to* OM*'s priorities, then the difference* $D$ *between the cache configurations of the two algorithms decreases by* 1, *i.e.,* $\Delta D = -1$. *Otherwise,* $\Delta D \leq 0$.

**Lemma D.2.** *Suppose that when serving a lazy adversary request, both* ONOPT *and* OM *miss. If* ONOPT *currently evicts a page according to* OM*'s priorities, then the difference* $D$ *between the cache configurations of the two algorithms does not increase, i.e.,* $\Delta D \leq 0$. *Otherwise,* $\Delta D \leq 1$.

$\qquad\square$

# E. Proof of Lemma D.1

**Lemma D.1.** *Suppose that when serving a lazy adversary request,* ONOPT *misses, whereas* OM *hits. If* ONOPT *currently evicts a page according to* OM*'s priorities, then the difference* $D$ *between the cache configurations of the two algorithms decreases by* 1, *i.e.,* $\Delta D = -1$. *Otherwise,* $\Delta D \leq 0$.

*Proof.* Let $\omega = (L_0, L_1, \ldots, L_k)$ be the current work function, $C$ be ONOPT's current cache configuration, and $C^*$ be that of OM. Since ONOPT misses and OM hits, the requested page $p$ satisfies $p \in C^* \setminus C$.

Denote $Q_j = L_1 \cup \cdots \cup L_j$. Let $p \in L_i$, and let $j \geq i$ be the smallest index such that $Q_j$ is saturated by $C$, meaning $|C \cap Q_j| = j$. By definition, $C \cap Q_j$ is ONOPT's eviction candidate set, and $p \in Q_j$. Recall that OM's cache $C^* = C_k^*(\omega)$ is constructed via the priority-greedy process (see Definition 2 of Brodal et al. (2015)), which imposes the validity constraint $|C^* \cap Q_l| \leq l$ for all $l$.

We consider two cases based on which eviction rule ONOPT applies.

1. ONOPT does not follow OM's priorities. Instead, it evicts an arbitrary page in $C \cap Q_j$. Since the evicted page may or may not belong to $C^*$, we have $\Delta D \leq 0$.

2. ONOPT follows OM's priorities. Below, we prove that the evicted page $q$ satisfies $q \notin C^*$, which implies $\Delta D = -1$.

   In this case, ONOPT evicts

   $$q = \arg\min_{x \in C \cap Q_j} \pi(x), \tag{9}$$

   where $\pi(x)$ denotes the priority of $x$ assigned by OM.

   Let $q \in L_b$, where $b \leq j$. Define

   $$h = \max\{0 \leq l < b : |C^* \cap Q_l| = l\},$$

   where $Q_0 = \emptyset$. Thus $Q_h$ is the last prefix before $q$'s layer that is saturated by $C^*$, and $Q_l$ is not saturated by $C^*$ for every $h < l < b$, i.e., $|C^* \cap Q_l| \leq l - 1$.

   **Claim.** There exists a page in $C \cap (Q_j \setminus Q_h)$ that is not in $C^*$, i.e., $(C \cap (Q_j \setminus Q_h)) \setminus C^* \neq \emptyset$.

*Proof of Claim.* Suppose for contradiction that $C \cap (Q_j \setminus Q_h) \subseteq C^*$. Since $C$ is a valid configuration, Corollary 3.2 gives $|C \cap Q_h| \leq h$. Combined with $|C \cap Q_j| = j$:

$$|C \cap (Q_j \setminus Q_h)| = j - |C \cap Q_h| \geq j - h.$$

Since $C \cap (Q_j \setminus Q_h) \subseteq C^*$ by assumption, and $|C^* \cap Q_h| = h$:

$$|C^* \cap Q_j| \geq h + |C \cap (Q_j \setminus Q_h)| \geq j.$$

By validity of $C^*$, $|C^* \cap Q_j| \leq j$. This forces $|C^* \cap Q_j| = j$, which in turn forces $|C \cap Q_h| = h$.

If $h \geq i$, then $Q_h$ is saturated by $C$ with $i \leq h < j$, contradicting the fact that $j$ is the smallest index no less than $i$ such that $Q_j$ is saturated by $C$. Hence $h < i \leq j$, so $p \in L_i \subseteq Q_j \setminus Q_h$. Now $C^* \cap Q_j$ contains: (i) the $h$ pages in $C^* \cap Q_h$, (ii) pages in $C \cap (Q_j \setminus Q_h) \subseteq C^*$, with at least $j - h$ pages, and (iii) the page $p \in (C^* \setminus C) \cap (Q_j \setminus Q_h)$. Therefore

$$|C^* \cap Q_j| \geq h + (j - h) + 1 = j + 1,$$

contradicting the validity of $C^*$. $\qquad\square$

Now let $y \in (C \cap (Q_j \setminus Q_h)) \setminus C^*$ be the page guaranteed by the claim.

Suppose, for contradiction, that $q \in C^*$. Since $y \notin C^*$ while $q \in C^*$, we have $q \neq y$. Since $q$ is the minimum-priority page in $C \cap Q_j$ by (9) and $y \in C \cap Q_j$, we have $\pi(q) < \pi(y)$.

Let $y \in L_a$. Since $y \in Q_j \setminus Q_h$, we have $h < a \leq j$. Consider the alternative configuration $\tilde{C}^* = C^* \setminus \{q\} \cup \{y\}$. We verify it satisfies the prefix constraints given by Corollary 3.2:

- If $a < b$: for each $a \leq l < b$, adding $y \in L_a$ increases $|\tilde{C}^* \cap Q_l|$ by 1 relative to $|C^* \cap Q_l|$, since $q \in L_b$ is not in $Q_l$. Because $h < a \leq l < b$ implies $|C^* \cap Q_l| \leq l - 1$, we retain $|\tilde{C}^* \cap Q_l| \leq l$. Moreover, all other prefixes are unchanged or decreased.
- If $a \geq b$: for each $b \leq l < a$, removing $q \in L_b$ decreases $|\tilde{C}^* \cap Q_l|$ by 1, and for $l \geq a$ and $l < b$, the count is unchanged. Thus, all constraints are preserved.

Hence, $\tilde{C}^*$ is still valid. Since $\pi(y) > \pi(q)$, the priority-greedy process of OM would select $y$ over $q$, contradicting $q \in C^*$ and $y \notin C^*$.

Therefore, $q \notin C^*$, and the eviction removes a page in $C \setminus C^*$, giving $\Delta D = -1$.

$\qquad\square$

## F. Proof of Lemma D.2

**Lemma D.2.** *Suppose that when serving a lazy adversary request, both* ONOPT *and* OM *miss. If* ONOPT *currently evicts a page according to* OM*'s priorities, then the difference $D$ between the cache configurations of the two algorithms does not increase, i.e., $\Delta D \leq 0$. Otherwise, $\Delta D \leq 1$.*

*Proof.* We adopt the notation from the proof of Lemma D.1 in Appendix E: $\omega$, $C$, $C^*$, $C_k^*(\omega)$, $Q_j$, and the eviction candidate set $C \cap Q_j$ with $j$ the smallest saturated index no less than $i$. The difference here is that both algorithms miss on $p$, i.e., $p \notin C$ and $p \notin C^*$. Similarly, we consider two cases based on which eviction rule ONOPT applies.

1. ONOPT does not follow OM's priorities. Instead, it evicts an arbitrary page in $C \cap Q_j$. Since this page may or may not belong to $C^*$, and OM may or may not evict a page belonging to $C$, we have $\Delta D \leq 1$.

2. ONOPT follows OM's priorities. Let $C'$ and $C^{*\prime}$ be the cache configurations of ONOPT and OM after serving $p$, respectively. Below, we prove that the evicted page $q$ satisfies $q \notin C^{*\prime}$.

   In this case, ONOPT evicts

   $$q = \arg \min_{x \in C \cap Q_j} \pi(x),$$

yielding $C' = C \setminus \{q\} \cup \{p\}$. Meanwhile, OM also loads $p$. By Definition 1 and Corollary 3.5, valid configurations after serving $p$ are exactly the valid configurations before serving $p$ that contain $p$, since $p$ incurs no cost. Therefore

$$C^{*\prime} = C_k^*(\omega^p)$$

is the priority-greedy valid configuration among all valid configurations containing $p$.

Since $p \in C^{*\prime} \setminus C$, we can replay the argument of Case 2 of Lemma D.1's proof with $C^{*\prime}$ in place of $C^*$. Specifically, let $q \in L_b$ with $b \leq j$, and define

$$h = \max\{0 \leq l < b : |C^{*\prime} \cap Q_l| = l\}$$

as before.

A similar claim holds: there exists a page $y \in (C \cap (Q_j \setminus Q_h)) \setminus C^{*\prime}$. The proof is identical to that in Lemma D.1, with the validity of $C^{*\prime}$ in place of $C^*$ and using the fact that $p \in (C^{*\prime} \setminus C) \cap (Q_j \setminus Q_h)$.

Suppose for contradiction $q \in C^{*\prime}$. By the same priority comparison, $\pi(q) < \pi(y)$. Consider the alternative configuration $\widetilde{C}^{*\prime} = C^{*\prime} \setminus \{q\} \cup \{y\}$. The validity check across the two cases $a < b$ and $a \geq b$ proceeds as before, so $\widetilde{C}^{*\prime}$ is a valid configuration. Since $\pi(y) > \pi(q)$, the priority-greedy process of OM would select $y$ over $q$, contradicting $q \in C^{*\prime}$ and $y \notin C^{*\prime}$. Therefore, $q \notin C^{*\prime}$.

Let $z$ be the page evicted by OM. If $z = q$, both algorithms perform the same, so $\Delta D = 0$.

Otherwise, $z \neq q$. Combined with $q \notin C^{*\prime}$, we have $q \notin C^*$. In this case, if $z \notin C$, both algorithms remove a page that is not in the cache configuration of the other, leading to $\Delta D = -1$. If $z \in C$, OM replaces $z \in C$ with $p \in C'$, and ONOPT replaces $q \notin C^*$ with $p \in C^{*\prime}$, so $\Delta D = 0$.

Therefore, $\Delta D \leq 0$ in this case.

$\square$

# G. Reanalysis of OM Algorithm

## G.1. Definition and Properties

Recall that OM's cache configuration $C = C_k(\omega)$ is constructed by the priority-greedy process in Definition 2 of Brodal et al. (2015):

$$\text{for } j = 1, ..., k, \quad C_j(\omega) = \max_j(C_{j-1}(\omega) \cup L_j) \tag{10}$$

where $C_0(\omega) = \emptyset$, and $\max_j(S)$ means the subset of $S$ of size $j$ having the largest priorities.

The following two conclusions are given in the literature.

**Theorem G.1** (Theorem 2 of Achlioptas et al. (2000)). *Algorithm* EQUITABLE *is* $H_k$-*competitive.*

**Theorem G.2** (Theorem 2 of Brodal et al. (2015)). *Assume that non-revealed pages are assigned priorities such that the order of the priorities is distributed uniformly at random. For any $\omega$, the distribution of $C_k$ over all possible cache configurations is the same as the distribution of the cache configurations for the* EQUITABLE *algorithms.*

**Lemma G.3** (Lemma 4 of Achlioptas et al. (2000)). *For any $\omega$, the expected cost of* EQUITABLE *is the same against all lazy adversary strategies for $\omega$.*

Theorem G.1 and Theorem G.2 together imply that OM is $H_k$-competitive. Theorem G.2 and Lemma G.3 imply the following corollary.

**Corollary G.4.** *For any $\omega$, the expected cost of* OM *is the same against all lazy adversary strategies for $\omega$.*

Moreover, the proof of Theorem G.1 of Achlioptas et al. (2000) further implies the following corollary, with the potential function $\phi(\omega)$ as defined in Definition 2.

**Corollary G.5** (Corollary 5.5 in the main text). *For any $\omega$, when* OM *serves a request to $p \in L_0$, the potential change satisfies $\Delta\phi = \phi(\omega^p) - \phi(\omega) \leq H_k - 1$.*

Building on the above, we establish a new lemma characterizing the potential change when serving lazy adversary requests.

### G.2. Potential Change for Lazy Adversary Requests

**Lemma 5.6.** *For any $\omega$, when* OM *serves a lazy adversary request to $p \in N(\omega)$, the potential change satisfies $\Delta\phi = \phi(\omega^p) - \phi(\omega) \leq -1/(U(\omega) + 1)$, where $U(\omega) = k - |R(\omega)|$ is the number of unrevealed layers.*

*Proof.* Let $C(\omega)$ denote the cache configuration of OM under work function $\omega$. The potential function $\phi(\omega)$ is defined in Definition 2. The following equation also follows from Corollary G.4.

$$\forall x \in N(\omega), \quad \phi(\omega) = \Pr[x \notin C(\omega)] + \phi(\omega^x). \tag{11}$$

By (11), we have

$$\Delta\phi = \phi(\omega^p) - \phi(\omega) = -\Pr[p \notin C(\omega)]. \tag{12}$$

Below, we proceed by induction on $U(\omega)$.

1. For the base case $U(\omega) = 1$, only one page in $N(\omega)$ exists in $C(\omega)$. Thus, there exists $y \in N(\omega)$ such that

$$\Pr[y \notin C(\omega)] \geq \frac{|N(\omega)| - 1}{|N(\omega)|} \geq \frac{U(\omega)}{U(\omega) + 1} \geq \frac{1}{2},$$

   where $|N(\omega)| \geq U(\omega) + 1 = 2$, as otherwise all pages in $N(\omega)$ are revealed. Moreover $\phi(\omega^p) = \phi(\omega^y) = 0$, as $U(\omega^p) = U(\omega^y) = 0$. Combining with (12),

$$\Delta\phi = \phi(\omega^p) - \phi(\omega) = \phi(\omega^y) - \phi(\omega) = -\Pr[y \notin C(\omega)] \leq -1/2.$$

2. Now we consider the case $U(\omega) > 1$, assuming that the lemma holds for any $\omega'$ with $U(\omega') = U(\omega) - 1$.

   Let $M_{total}(\omega)$ denote the total miss probability mass over unrevealed pages. Since all pages in $R(\omega)$ reside in the cache, exactly $U(\omega)$ pages in $N(\omega)$ exist in $C(\omega)$, and

$$M_{total}(\omega) = \sum_{x \in N(\omega)} \Pr[x \notin C(\omega)] = |N(\omega)| - U(\omega).$$

   After serving $p$,

$$M_{total}(\omega^p) = \sum_{x \in N(\omega^p)} \Pr[x \notin C(\omega^p)] = |N(\omega^p)| - U(\omega^p).$$

   Since $p$ leaves $N(\omega)$ and the number of unrevealed layers decreases by 1, $p$ itself does not change $M_{total}$. Moreover, $p$ then enters the cache with certainty, i.e., $\Pr[p \notin C(\omega^p)] = 0$, so its original miss probability $\Pr[p \notin C(\omega)]$ is redistributed among the remaining unrevealed pages in $N(\omega^p)$.

   Moreover, if $p \in L_1$ and there are multiple pages in $L_1$, the remaining pages in $L_1$ leave $S(\omega)$ upon $p$'s request. Denote this set by $L = N(\omega) \backslash N(\omega^p) \backslash \{p\}$. Their departure reduces the number of unrevealed pages without changing the number of unrevealed layers, decreasing $M_{total}$ by $|L|$. Since their total original miss probability mass satisfies $\sum_{x \in L} \Pr[x \notin C(\omega)] \leq |L|$, the total miss probability mass of remaining pages in $N(\omega^p)$ can only decrease.

   Therefore, there exists $y \in N(\omega^p)$ such that

$$\Delta_y = \Pr[y \notin C(\omega^p)] - \Pr[y \notin C(\omega)] \leq \frac{\Pr[p \notin C(\omega)]}{|N(\omega^p)|}. \tag{13}$$

   Since $y \in N(\omega^p) \subset N(\omega)$, applying (11) to $\omega$ and $\omega^p$ gives

$$\phi(\omega) = \Pr[y \notin C(\omega)] + \phi(\omega^y) \quad \text{and} \quad \phi(\omega^p) = \Pr[y \notin C(\omega^p)] + \phi(\omega^{\langle p, y \rangle}).$$

Combining these two equations, we have

$$\Delta\phi = \phi(\omega^p) - \phi(\omega) = \Delta_y + \phi(\omega^{\langle p,y \rangle}) - \phi(\omega^p)$$

$$\leq \frac{\Pr[p \notin C(\omega)]}{|N(\omega^p)|} + \phi(\omega^{\langle p,y \rangle}) - \phi(\omega^p) \quad \text{by (13)}$$

$$= -\frac{\Delta\phi}{|N(\omega^p)|} + \phi(\omega^{\langle p,y \rangle}) - \phi(\omega^p) \quad \text{by (12)}.$$

This gives

$$\Delta\phi \leq (\phi(\omega^{\langle p,y \rangle}) - \phi(\omega^p)) \cdot \frac{|N(\omega^p)|}{1 + |N(\omega^p)|}.$$

Since $U(\omega^p) = U(\omega) - 1$ and $y \in N(\omega^p)$, by the inductive hypothesis,

$$\phi(\omega^{\langle p,y \rangle}) - \phi(\omega^p) \ \leq \ -\frac{1}{U(\omega^p) + 1} = -\frac{1}{U(\omega)}.$$

Finally, by $|N(\omega^p)| \geq U(\omega^p) + 1 = U(\omega)$,

$$\Delta\phi \leq -\frac{1}{U(\omega)} \cdot \frac{|N(\omega^p)|}{1 + |N(\omega^p)|} \leq -\frac{1}{U(\omega)} \cdot \frac{U(\omega)}{U(\omega) + 1} = -\frac{1}{U(\omega) + 1}.$$

This completes the proof.

$\square$

# H. RPB-OM

Algorithm 3 describes RPB-OM, which plugs OM's priorities into RPB-ONOPT.

## H.1. Smoothness of RPB-OM

To prove smoothness, we define a new active residual potential to replace the original $\phi(\omega)$ in the potential of RPB-OM. This allows us to focus on the impact of prediction errors.

**(1) The definition of active residual potential.** Fix a request sequence and a realization of the priority function $\pi$. Each $L_0$-miss creates one residual event $e$. Let $\omega_e$ be the work function immediately before event $e$, and let $C(\omega_e)$ be the cache configuration of RPB-OM under $\omega_e$. Since the request is to a page $p \in L_0$, the eviction-candidate set is the whole cache configuration, i.e., $V_e = C(\omega_e)$.

By definition, RPB-OM evicts

$$q_e = \arg\max_{x \in V_e} \tilde{\ell}_e(x),$$

where $\tilde{\ell}_e(x)$ is the predicted next-arrival time of page $x$ at event $e$. Let $\ell_e(x)$ be the true next-arrival time of $x$ after event $e$.

Define the local inversion of event $e$ by

$$\text{inv}_e = |\{x \in V_e \setminus \{q_e\} : \ell_e(q_e) < \ell_e(x)\}|.$$

The event $e$ initially tracks a residual set

$$Z_e(\omega_e^p) = \{q_e\} \cup \{x \in V_e \setminus \{q_e\} : \ell_e(q_e) < \ell_e(x)\}.$$

At a work function $\omega$, $Z_e(\omega)$ consists of the pages in this initial set that have not yet been accounted for. Here, a page is accounted for once it is requested, or once it leaves the unrevealed layers due to lazy-layer updates.

After being accounted for, the page leaves the residual set. Let

$$z_e(\omega_e^p) = |Z_e(\omega_e^p)|.$$

---

**Algorithm 3** RPB-OM

---

1: $\omega :=$ the current work function
2: $C :=$ the current cache configuration
3: $B :=$ the prediction budget
4: $\tau :=$ the $O(1)$ budget value reset at an $L_0$-miss
5: $Y :=$ the eviction-effectiveness metric
6: $V :=$ the eviction candidate set
7: **while** receiving a request to page $p$ **do**
8:    **if** $p \notin C$ **then**
9:       **if** $p \in L_0$ **then**
10:          Evict a page $x \in C$ following the predictions
11:          $B \leftarrow \tau$
12:       **else if** $p \in L_i, i > 0$ **then**
13:          Apply OM's rule to select eviction candidates: $V \leftarrow C \cap \bigcup_{l=1}^{z} L_l$, where $z = \min\Big\{ j \in \{i,..,k\} \mid$
         $|C \cap \bigcup_{l=1}^{j} L_l| = j\Big\}$
14:          **if** $U(\omega) \leq (Y+2)/e - 2$ **then**
15:             $B \leftarrow B + 1$
16:          **if** $B > 0$ **then**
17:             Evict a page $x \in V$ following the predictions
18:             $B \leftarrow B - 1$
19:          **else**
20:             Evict a page $x \in V$ with the lowest OM priority
21:       Cache page $p$, update $C$
22:       $Y \leftarrow U(\omega)$

---

The event $e$ is active if $z_e(\omega) > 0$, and we denote the set of active events by $\mathcal{A}(\omega)$.

Consider some event $e'$ with an initial residual set $Z_{e'}$. A page is removed from $Z_{e'}$ once it is accounted for, namely once it is requested or once it leaves the unrevealed layers due to lazy-layer updates. Once $Z_{e'}$ becomes empty, the event $e'$ becomes inactive. Hence, each miss caused by a lazy adversary request is covered by at least one active residual event.

For a lazy adversary request to $p$ at work function $\omega$, define

$$X_p(\omega) = \{e \in \mathcal{A}(\omega) : p \in Z_e(\omega)\}$$

as the set of active residual events that cover this request.

Finally, we define the active residual potential as

$$R^{act}(\omega) = \sum_{e \in \mathcal{A}(\omega)} H_{z_e(\omega)}.$$

**(2) A new potential $\Psi(\omega)$.** Let $C(\omega)$ be the cache configuration of RPB-OM under $\omega$, and let $C^*(\omega)$ be the cache configuration of OM under the same priority function $\pi$. The new potential of RPB-OM is

$$\Psi(\omega) = R^{\mathrm{act}}(\omega) + D(\omega) + B(\omega),$$

where $D(\omega)$ and $B(\omega)$ use the same definitions as in Section 5.4, representing the cache difference between RPB-OM and OM, and the remaining prediction budget, respectively.

**(3) Potential change upon lazy requests.** We first prove Lemma H.1 below, which shows that when serving a lazy adversary request, the change in $R^{act}$ is upper bounded by the corresponding change in OM's potential $\phi$.

**Lemma H.1.** *For a lazy adversary request to page $p$ at work function $\omega$, we have*

$$R^{\mathrm{act}}(\omega^p) - R^{\mathrm{act}}(\omega) \leq \phi(\omega^p) - \phi(\omega),$$

where $\phi$ is OM's potential as defined in Definition 2.

*Proof.* We first bound the miss probability of OM.

Let $C^*(\omega)$ be the cache configuration of OM under $\omega$. Since OM maintains the same cache-configuration distribution as EQUITABLE, the only randomness relevant to whether $p \in C^*(\omega)$ is the relative priority order among the pages in the residual sets.

For each active residual event $e$, the set $Z_e(\omega)$ is determined by the event that created $e$ and by the subsequent work-function evolution. That is to say, its update is independent of the algorithm. Hence, conditioned on $Z_e(\omega)$, the relative priority order of pages inside $Z_e(\omega)$ is still uniform.

The residual event itself makes one page in $Z_e(\omega)$ absent from the cache. By the priority symmetry of OM, for every $e \in X_p(\omega)$,

$$\Pr[p = \min_\pi(Z_e(\omega))] = \frac{1}{z_e(\omega)},$$

where $\min_\pi(Z_e(\omega))$ denotes the page with the minimal priority assigned by OM in $Z_e(\omega)$.

Each miss caused by a lazy adversary request is covered by some active residual events whose residual set contains the requested page. Therefore, the miss probability of OM on $p$ is

$$x_p(\omega) \leq \sum_{e \in X_p(\omega)} \Pr[p = \min_\pi(Z_e(\omega))] = \sum_{e \in X_p(\omega)} \frac{1}{z_e(\omega)}. \tag{14}$$

Now consider the change of $R^{\mathrm{act}}$.

Moreover, if $e \in X_p(\omega)$, then $p \in Z_e(\omega)$. After serving the request to $p$, the page $p$ is revealed and is removed from every such residual candidate set. Therefore,

$$z_e(\omega^p) \leq z_e(\omega) - 1.$$

Hence,

$$H_{z_e(\omega^p)} - H_{z_e(\omega)} \leq H_{z_e(\omega)-1} - H_{z_e(\omega)} = -\frac{1}{z_e(\omega)}.$$

Since no new residual event is created by the current request, and all other active events contribute at most zero to the change, summing over all active events gives

$$R^{\mathrm{act}}(\omega^p) - R^{\mathrm{act}}(\omega) \leq \sum_{e \in X_p(\omega)} H_{z_e(\omega^p)} - H_{z_e(\omega)}$$

$$\leq - \sum_{e \in X_p(\omega)} \frac{1}{z_e(\omega)}$$

$$\leq -x_p(\omega) \quad \text{by (14)}.$$

Finally, by $\phi(\omega^p) - \phi(\omega) = -x_p(\omega)$, we have

$$R^{\mathrm{act}}(\omega^p) - R^{\mathrm{act}}(\omega) \leq \phi(\omega^p) - \phi(\omega).$$

This completes the proof. $\qquad\square$

**Lemma H.2.** *For a lazy request to $p \in S(\omega)$ when the work function is $\omega$, we have*

$$\Delta cost(\text{RPB-OM}) + \Delta\Psi = \Delta cost(\text{RPB-OM}) + \Psi(\omega^p) - \Psi(\omega) \leq 0. \tag{15}$$

*Proof.* If the lazy request is to a revealed page, both RPB-OM and OM hit, and no unrevealed page is revealed. Hence, $\Delta R^{act} = \Delta D = \Delta B = 0$, and (15) holds.

Otherwise, the lazy request is to an unrevealed page, i.e., a lazy adversary request. If RPB-OM hits, then $\Delta R^{act} \leq 0$, $\Delta D \leq 0$ and $\Delta B = 0$, so (15) holds. For the case where RPB-OM misses, we bound $\Delta R^{act}$. By Lemma H.1, we have

$$R^{\text{act}}(\omega^p) - R^{\text{act}}(\omega) \leq \phi(\omega^p) - \phi(\omega).$$

Therefore, $\Delta R^{act} \leq \Delta\phi$. Using the same argument as in the proof of Theorem 5.7, with $\Delta\phi$ replaced by $\Delta R^{act}$, we obtain

$$\Delta cost(\text{RPB-OM}) + \Delta\Psi \leq 0. \tag{16}$$

This completes the proof. □

**(4) Potential change upon $L_0$-misses.**

**Lemma H.3.** *Consider a request to $p \in L_0$ when the work function is $\omega$. We have*

$$\Delta\Psi = \Psi(\omega^p) - \Psi(\omega) \leq H_{\text{inv}} + O(1).$$

*Proof.* We decompose $\Delta\Psi = \Delta R^{act} + \Delta D + \Delta B$.

First, we bound $\Delta R^{act}$. We split it into the contribution of old events and the contribution of the event newly created by the current prediction-driven eviction.

Consider an event $e'$ created before the request. By definition, the residual set $Z_{e'}(\omega)$ consists only of pages from the eviction candidate set at the time $e'$ was created. The current request is to $p \in L_0$, and the $L_0$-miss update does not add any new page to the original eviction candidate set of $e'$. Moreover, a page can only be removed from $Z_{e'}$ when it is *accounted for*, that is, when it is requested or when it leaves the unrevealed layers due to lazy-layer updates. Hence,

$$Z_{e'}(\omega^p) \subseteq Z_{e'}(\omega) \quad \text{and} \quad z_{e'}(\omega^p) \leq z_{e'}(\omega).$$

Therefore, the total contribution of old events to $\Delta R^{act}$ is at most 0.

Now consider the newly created event. The current prediction-driven eviction evicts $q$ from $V = C(\omega)$. Let

$$I(\omega) = \{x \in V \setminus \{q\} : \ell(q) < \ell(x)\},$$

be the set of pages with larger next-arrival times than $q$. Then, $|I(\omega)|$ equals the local inversion $\text{inv}$.

All pages in $V \setminus (\{q\} \cup I(\omega))$ have true next-arrival time earlier than $q$. Thus, before $q$ returns, these pages will have either been requested or removed from the unrevealed layers. Consequently, after the first return of $q$, the pages in the original eviction candidate set $V$ that can still remain unaccounted are contained in $I(\omega)$. Hence, the initial residual scale of the new event is

$$z(\omega^p) = |Z(\omega^p)| = |\{q\} \cup I(\omega)| = \text{inv} + 1.$$

Therefore, the potential added by the new event is $H_{z(\omega^p)} = H_{\text{inv}+1}$. Combining the fact that old events do not increase this potential, we obtain

$$\Delta R^{act} = R^{\text{act}}(\omega^p) - R^{\text{act}}(\omega) \leq H_{\text{inv}+1}.$$

Second, since both RPB-OM and OM miss upon the request to $p \in L_0$, $\Delta D \leq 1$. Moreover, $\Delta B \leq \tau$ in this case (since $B$ is reset to $\tau$).

Combining these inequalities gives

$$\begin{aligned}
\Delta\Psi = \Delta R^{act} + \Delta D + \Delta B \\
\leq H_{\text{inv}+1} + 1 + \tau \\
\leq H_{\text{inv}} + 2 + \tau \\
= H_{\text{inv}} + O(1),
\end{aligned}$$

where $H_0 = 0$. This completes the proof. □

**(5) The proof of smoothness.**

**Theorem 5.8.** RPB-OM *is* $O(1 + \log(\eta_1/cost(\text{OPT})))$-*smooth, where* $\eta_1$ *denotes the total* $\ell_1$ *error in the predicted next-arrival times.*

*Proof.* Let $\mathcal{E}$ be the set of events created upon $L_0$-misses. We first derive a global amortized bound.

By Lemma H.3, each event $e \in \mathcal{E}$ increases the potential by at most

$$H_{\text{inv}_e} + O(1).$$

The actual miss cost of this request contributes an additional 1, which is absorbed into the $O(1)$ term.

By Lemma H.2, whenever RPB-OM misses on a lazy adversary request, the expected potential decreases by at least 1. Hence the miss cost on such requests is paid by the potential decrease. If RPB-OM hits, the request incurs no cost and the potential does not increase. Therefore, the lazy adversary requests contribute no positive amortized cost.

Since the potential is nonnegative and initially zero, summing up gives

$$cost(\text{RPB-OM}) \leq O(|\mathcal{E}|) + \sum_{e \in \mathcal{E}} H_{\text{inv}_e}.$$

Moreover, we have

$$cost(\text{OPT}) \geq |\mathcal{E}|.$$

Thus

$$cost(\text{RPB-OM}) \leq O(1) \cdot cost(\text{OPT}) + \sum_{e \in \mathcal{E}} H_{\text{inv}_e}.$$

By Lemma 4.1 of Rohatgi (2020), the total number of inversions created by prediction-driven $L_0$-misses satisfies

$$\sum_{e \in \mathcal{E}} \text{inv}_e \leq 2\eta_1.$$

Finally, by the concavity of $H_x$ and Jensen's inequality, we have

$$\frac{cost(\text{RPB-OM})}{cost(\text{OPT})} \leq O\Big(1 + \log\Big(\frac{\eta_1}{cost(\text{OPT})}\Big)\Big),$$

$\square$

# I. Additional Experimental Results

For brevity, we use the following abbreviations: BO for BlindOracle, BO&L for BlindOracle&LRU, OM for OnlineMin, LM for LMarker, PM for PredictiveMarker, OOM for OnOPT-OM, and ROM-$x$ for RPB-OM with $\tau = x$. We highlight in bold the best performance among all algorithms (excluding OPT) for each trace.

### I.1. Detailed Results with the PLECO and POPU Predictors

Table 2 and Table 3 report per-trace results on SPEC CPU 2006 with the PLECO predictor. We also evaluate RPB-OM at additional values of $\tau$. Experimental results show that RPB-OM achieves the best average performance while remaining robust when BLINDORACLE performs worse than classical heuristics.

*Table 2.* Hit Ratio (%) of algorithms with the PLECO predictor across SPEC CPU 2006 datasets. We highlight in bold the best performance among all algorithms (excluding OPT) for each trace.

| Algorithm | astar | bwaves | bzip | cactus | gems | lbm | les3d | libq | mcf | milc | omnet | sph3 | xalanc | Mean | Std |
|---|---|---|---|---|---|---|---|---|---|---|---|---|---|---|---|
| OPT | 37.39 | 4.90 | 80.81 | 33.69 | 12.21 | 24.83 | 30.88 | 5.30 | 44.56 | 1.38 | 42.43 | 74.73 | 56.89 | 34.62 | 25.60 |
| LRU | 4.01 | 0.00 | **63.81** | 0.00 | 2.88 | 0.00 | 9.49 | 0.00 | 27.09 | 0.01 | **20.44** | 12.74 | **45.08** | 14.27 | 20.17 |
| Marker | 4.82 | 0.00 | 63.57 | 1.19 | 4.04 | 0.02 | 9.41 | 0.00 | 25.64 | 0.02 | 20.32 | 42.25 | 43.31 | 16.51 | 21.13 |
| OM | 8.68 | 0.02 | 61.15 | 6.11 | 3.90 | 1.83 | 9.66 | 0.00 | 19.07 | 0.01 | 15.84 | 48.72 | 36.90 | 16.30 | 20.12 |
| BO | **23.16** | 0.07 | 51.54 | 3.87 | 0.35 | 0.47 | 6.95 | **5.30** | 9.09 | **1.07** | 9.84 | 66.44 | 28.75 | 15.92 | 21.24 |
| BO&L | **23.16** | 0.07 | 63.29 | 3.87 | 2.72 | 0.47 | 9.91 | **5.30** | 27.08 | **1.07** | 19.92 | 66.42 | 43.56 | 20.53 | 23.55 |
| PM | 5.52 | 0.00 | 62.01 | 2.99 | **4.69** | 0.12 | 9.49 | 0.00 | 26.00 | 0.02 | 20.23 | 62.19 | 42.07 | 18.10 | 23.20 |
| LM | 5.54 | 0.00 | 62.22 | 2.99 | **4.69** | 0.12 | 9.45 | 0.00 | 26.01 | 0.02 | 20.32 | 61.61 | 42.48 | 18.11 | 23.18 |
| F&R | 5.99 | **0.17** | 61.35 | 3.03 | 4.17 | 0.63 | 10.54 | **5.30** | 25.28 | 0.71 | 19.99 | 52.72 | 43.06 | 17.92 | 21.34 |
| OOM | 15.29 | 0.12 | 62.13 | 15.41 | 1.22 | 10.84 | 16.77 | **5.30** | 35.35 | 0.75 | 15.80 | 71.85 | 40.58 | 22.42 | 23.34 |
| ROM-1 | 16.71 | 0.12 | 61.58 | 17.27 | 1.21 | 10.84 | **16.82** | **5.30** | 37.27 | 0.77 | 15.84 | 72.19 | 39.77 | **22.75** | 23.29 |
| ROM-2 | 17.07 | 0.12 | 61.11 | 17.43 | 1.23 | 10.83 | 16.81 | **5.30** | 37.39 | 0.77 | 15.85 | **72.24** | 38.99 | 22.70 | 23.18 |
| ROM-4 | 17.19 | 0.12 | 60.77 | **17.46** | 1.22 | 10.83 | 16.81 | **5.30** | **37.40** | 0.77 | 15.78 | **72.24** | 38.40 | 22.64 | 23.10 |
| ROM-8 | 17.23 | 0.12 | 60.75 | **17.46** | 1.22 | 10.83 | 16.81 | **5.30** | **37.40** | 0.77 | 15.78 | **72.24** | 38.18 | 22.62 | 23.08 |
| ROM-16 | 17.23 | 0.12 | 60.75 | **17.46** | 1.22 | 10.83 | 16.81 | **5.30** | **37.40** | 0.77 | 15.78 | **72.24** | 38.18 | 22.62 | 23.08 |

*Table 3.* Cost Ratio of all algorithms with the PLECO predictor across SPEC CPU 2006 datasets. We highlight in bold the best performance among all algorithms (excluding OPT) for each trace.

| Algorithm | astar | bwaves | bzip | cactus | gems | lbm | les3d | libq | mcf | milc | omnet | sph3 | xalanc | Mean | Std |
|---|---|---|---|---|---|---|---|---|---|---|---|---|---|---|---|
| OPT | 1.000 | 1.000 | 1.000 | 1.000 | 1.000 | 1.000 | 1.000 | 1.000 | 1.000 | 1.000 | 1.000 | 1.000 | 1.000 | 1.000 | 0.000 |
| LRU | 1.533 | 1.051 | **1.886** | 1.508 | 1.106 | 1.330 | 1.309 | 1.056 | 1.315 | 1.014 | **1.382** | 3.453 | **1.274** | 1.478 | 0.640 |
| Marker | 1.520 | 1.051 | 1.899 | 1.490 | 1.093 | 1.330 | 1.311 | 1.056 | 1.341 | 1.014 | 1.384 | 2.286 | 1.315 | 1.392 | 0.361 |
| OM | 1.459 | 1.051 | 2.025 | 1.416 | 1.095 | 1.306 | 1.307 | 1.056 | 1.460 | 1.014 | 1.462 | 2.029 | 1.464 | 1.396 | 0.329 |
| BO | **1.227** | 1.051 | 2.525 | 1.450 | 1.135 | 1.324 | 1.346 | **1.000** | 1.640 | **1.003** | 1.566 | 1.328 | 1.653 | 1.404 | 0.405 |
| BO&L | **1.227** | 1.051 | 1.913 | 1.450 | 1.108 | 1.324 | 1.303 | **1.000** | 1.315 | **1.003** | 1.391 | 1.329 | 1.309 | 1.286 | 0.239 |
| PM | 1.509 | 1.051 | 1.980 | 1.463 | **1.086** | 1.329 | 1.309 | 1.056 | 1.335 | 1.014 | 1.386 | 1.496 | 1.344 | 1.335 | 0.261 |
| LM | 1.509 | 1.051 | 1.969 | 1.463 | **1.086** | 1.329 | 1.310 | 1.056 | 1.335 | 1.014 | 1.384 | 1.519 | 1.334 | 1.335 | 0.260 |
| F&R | 1.502 | **1.050** | 2.014 | 1.462 | 1.092 | 1.322 | 1.294 | **1.000** | 1.348 | 1.007 | 1.390 | 1.871 | 1.321 | 1.359 | 0.310 |
| OOM | 1.353 | **1.050** | 1.974 | 1.276 | 1.125 | **1.186** | 1.204 | **1.000** | 1.166 | 1.006 | 1.462 | 1.114 | 1.378 | 1.253 | 0.259 |
| ROM-1 | 1.330 | **1.050** | 2.002 | 1.248 | 1.125 | **1.186** | 1.203 | **1.000** | 1.131 | 1.006 | 1.462 | 1.101 | 1.397 | **1.249** | 0.268 |
| ROM-2 | 1.325 | **1.050** | 2.027 | **1.245** | 1.125 | **1.186** | 1.204 | **1.000** | **1.129** | 1.006 | 1.462 | **1.099** | 1.415 | 1.252 | 0.274 |
| ROM-4 | 1.323 | **1.050** | 2.045 | **1.245** | 1.125 | **1.186** | 1.203 | **1.000** | **1.129** | 1.006 | 1.463 | **1.099** | 1.429 | 1.254 | 0.279 |
| ROM-8 | 1.322 | **1.050** | 2.045 | **1.245** | 1.125 | **1.186** | 1.203 | **1.000** | **1.129** | 1.006 | 1.463 | **1.099** | 1.434 | 1.254 | 0.280 |
| ROM-16 | 1.322 | **1.050** | 2.045 | **1.245** | 1.125 | **1.186** | 1.203 | **1.000** | **1.129** | 1.006 | 1.463 | **1.099** | 1.434 | 1.254 | 0.280 |

Similarly, Table 4 and Table 5 report per-trace results on the SPEC CPU 2006 benchmark using the POPU predictor.

*Table 4.* Hit Ratio (%) of algorithms with the POPU predictor across SPEC CPU 2006 datasets. We highlight in bold the best performance among all algorithms (excluding OPT) for each trace.

| Algorithm | astar | bwaves | bzip | cactus | gems | lbm | les3d | libq | mcf | milc | omnet | sph3 | xalanc | Mean | Std |
|---|---|---|---|---|---|---|---|---|---|---|---|---|---|---|---|
| OPT | 37.39 | 4.90 | 80.81 | 33.69 | 12.21 | 24.83 | 30.88 | 5.30 | 44.56 | 1.38 | 42.43 | 74.73 | 56.89 | 34.62 | 25.60 |
| LRU | 4.01 | 0.00 | 63.81 | 0.00 | 2.88 | 0.00 | 9.49 | 0.00 | 27.09 | 0.01 | 20.44 | 12.74 | **45.08** | 14.27 | 20.17 |
| Marker | 4.72 | 0.00 | 63.25 | 1.23 | 4.06 | 0.03 | 9.38 | 0.00 | 25.67 | 0.02 | 20.38 | 42.23 | 43.34 | 16.49 | 21.08 |
| OM | 8.76 | 0.02 | 60.41 | 5.83 | 3.80 | 1.83 | 9.78 | 0.00 | 19.38 | 0.01 | 15.77 | 48.96 | 36.96 | 16.27 | 20.04 |
| BO | **33.18** | 0.11 | 63.24 | 13.98 | 0.38 | 0.47 | 10.84 | **5.30** | 29.75 | **1.07** | 16.81 | **71.96** | 35.61 | 21.75 | 23.99 |
| BO&L | **33.18** | 0.11 | 65.25 | 13.98 | 2.73 | 0.47 | 10.42 | **5.30** | 32.11 | **1.07** | 19.74 | 71.95 | 43.97 | 23.10 | 24.65 |
| PM | 6.81 | 0.00 | 65.14 | 3.38 | 4.14 | 0.09 | 11.02 | 0.00 | 31.46 | 0.02 | **22.19** | 59.50 | 45.00 | 19.13 | 23.68 |
| LM | 6.83 | 0.00 | 64.99 | 3.38 | 4.14 | 0.09 | 11.02 | 0.00 | 31.45 | 0.02 | 22.09 | 57.26 | 44.88 | 18.93 | 23.33 |
| F&R | 10.19 | 0.18 | 63.49 | 4.28 | **4.16** | 0.63 | 11.78 | 1.32 | 30.87 | 0.71 | 20.95 | 58.43 | 43.72 | 19.29 | 22.72 |
| OOM | 19.91 | **2.40** | 65.43 | 25.12 | 0.93 | 3.44 | **17.95** | 3.76 | 43.75 | 0.69 | 20.10 | 61.39 | 44.03 | 23.76 | 23.00 |
| ROM-1 | 20.25 | **2.40** | 65.72 | 27.27 | 0.96 | 3.36 | 17.64 | 4.30 | **43.77** | 0.68 | 20.61 | 65.62 | 43.83 | 24.34 | 23.61 |
| ROM-2 | 20.23 | **2.40** | 65.70 | **27.39** | 0.96 | 3.27 | 17.59 | 4.57 | **43.77** | 0.68 | 20.65 | 67.31 | 43.83 | 24.49 | 23.84 |
| ROM-4 | 20.23 | **2.40** | 65.82 | **27.39** | 0.97 | 3.24 | 17.58 | 4.77 | **43.77** | 0.68 | 20.68 | 67.95 | 43.61 | **24.55** | 23.93 |
| ROM-8 | 20.23 | **2.40** | **65.85** | **27.39** | 0.97 | 3.24 | 17.58 | 4.77 | **43.77** | 0.68 | 20.67 | 68.02 | 43.59 | **24.55** | 23.94 |
| ROM-16 | 20.23 | **2.40** | **65.85** | **27.39** | 0.97 | 3.24 | 17.58 | 4.77 | **43.77** | 0.68 | 20.66 | 68.02 | 43.60 | **24.55** | 23.94 |

*Table 5.* Cost Ratio of algorithms with the POPU predictor across SPEC CPU 2006 datasets. We highlight in bold the best performance among all algorithms (excluding OPT) for each trace.

| Algorithm | astar | bwaves | bzip | cactus | gems | lbm | les3d | libq | mcf | milc | omnet | sph3 | xalanc | Mean | Std |
|---|---|---|---|---|---|---|---|---|---|---|---|---|---|---|---|
| OPT | 1.000 | 1.000 | 1.000 | 1.000 | 1.000 | 1.000 | 1.000 | 1.000 | 1.000 | 1.000 | 1.000 | 1.000 | 1.000 | 1.000 | 0.000 |
| LRU | 1.533 | 1.051 | 1.886 | 1.508 | 1.106 | 1.330 | 1.309 | 1.056 | 1.315 | 1.014 | 1.382 | 3.453 | **1.274** | 1.478 | 0.640 |
| Marker | 1.522 | 1.051 | 1.915 | 1.490 | 1.093 | 1.330 | 1.311 | 1.056 | 1.341 | 1.014 | 1.383 | 2.286 | 1.314 | 1.393 | 0.363 |
| OM | 1.457 | 1.051 | 2.063 | 1.420 | 1.096 | 1.306 | 1.305 | 1.056 | 1.454 | 1.014 | 1.463 | 2.020 | 1.462 | 1.397 | 0.334 |
| BO | **1.067** | 1.050 | 1.915 | 1.297 | 1.135 | 1.324 | 1.290 | **1.000** | 1.267 | **1.003** | 1.445 | **1.110** | 1.493 | 1.261 | 0.254 |
| BO&L | **1.067** | 1.050 | 1.811 | 1.297 | 1.108 | 1.324 | 1.296 | **1.000** | 1.224 | **1.003** | 1.394 | **1.110** | 1.300 | 1.230 | 0.220 |
| PM | 1.489 | 1.051 | 1.817 | 1.457 | **1.092** | 1.329 | 1.287 | 1.056 | 1.236 | 1.014 | **1.351** | 1.603 | 1.276 | 1.312 | 0.237 |
| LM | 1.488 | 1.051 | 1.825 | 1.457 | **1.092** | 1.329 | 1.287 | 1.056 | 1.236 | 1.014 | 1.353 | 1.692 | 1.278 | 1.320 | 0.248 |
| F&R | 1.434 | 1.050 | 1.903 | 1.444 | **1.092** | 1.322 | 1.276 | 1.042 | 1.247 | 1.007 | 1.373 | 1.645 | 1.306 | 1.319 | 0.256 |
| OOM | 1.279 | **1.026** | 1.801 | 1.129 | 1.128 | 1.285 | **1.187** | 1.016 | 1.015 | 1.007 | 1.388 | 1.528 | 1.298 | 1.237 | 0.234 |
| ROM-1 | 1.274 | **1.026** | 1.786 | 1.097 | 1.128 | 1.286 | 1.192 | 1.010 | **1.014** | 1.007 | 1.379 | 1.361 | 1.303 | 1.220 | 0.219 |
| ROM-2 | 1.274 | **1.026** | 1.787 | **1.095** | 1.128 | 1.287 | 1.192 | 1.008 | **1.014** | 1.007 | 1.378 | 1.294 | 1.303 | 1.215 | 0.216 |
| ROM-4 | 1.274 | **1.026** | 1.781 | **1.095** | 1.128 | 1.287 | 1.192 | 1.006 | **1.014** | 1.007 | 1.378 | 1.268 | 1.308 | 1.213 | 0.215 |
| ROM-8 | 1.274 | **1.026** | **1.779** | **1.095** | 1.128 | 1.287 | 1.192 | 1.006 | **1.014** | 1.007 | 1.378 | 1.266 | 1.308 | **1.212** | 0.214 |
| ROM-16 | 1.274 | **1.026** | 1.780 | **1.095** | 1.128 | 1.287 | 1.192 | 1.006 | **1.014** | 1.007 | 1.378 | 1.266 | 1.308 | **1.212** | 0.214 |

# J. Sensitivity Analysis of $\tau$

The cache size $k$ is 16 when evaluating on the SPEC CPU 2006 benchmark, as the associativity is set to 16, which is standard in the literature. This limits the range of $\tau$ and the sensitivity we can evaluate. Therefore, we conduct additional experiments on the commonly used BrightKite (Cho et al., 2011) and CitiBike (CitiBike) datasets with $k = 1000$ to better evaluate the sensitivity of $\tau$. We vary $\tau$ from 0 to 1000 and report the hit ratio and cost ratio in Tables 6 and 7.

*Table 6.* Sensitivity analysis of $\tau$ on the BrightKite dataset ($k = 1000$). Hit ratio (HR) and cost ratio (CR) of RPB-OM under varying $\tau$ and log-normal noise parameter $\sigma$.

| $\tau$ | Hit Ratio | | | | | Cost Ratio $\downarrow$ | | | | |
|---|---|---|---|---|---|---|---|---|---|---|
| | $\sigma=5$ | $\sigma=10$ | $\sigma=20$ | $\sigma=30$ | $\sigma=50$ | $\sigma=5$ | $\sigma=10$ | $\sigma=20$ | $\sigma=30$ | $\sigma=50$ |
| 1 | 0.819 | 0.798 | 0.785 | 0.781 | 0.778 | 1.118 | 1.246 | 1.327 | 1.353 | 1.372 |
| 2 | **0.820** | 0.800 | 0.788 | 0.785 | 0.781 | 1.115 | 1.236 | 1.309 | 1.330 | 1.350 |
| 5 | **0.820** | **0.801** | 0.789 | 0.785 | 0.782 | **1.113** | 1.229 | 1.303 | 1.328 | 1.347 |
| 10 | **0.820** | **0.801** | **0.790** | **0.786** | **0.783** | 1.114 | 1.228 | 1.300 | **1.323** | 1.343 |
| 20 | **0.820** | **0.801** | 0.789 | 0.785 | **0.783** | 1.115 | 1.228 | 1.302 | 1.325 | **1.340** |
| 50 | **0.820** | **0.801** | 0.789 | 0.785 | **0.783** | 1.114 | 1.231 | 1.302 | 1.327 | 1.342 |
| 100 | **0.820** | **0.801** | 0.789 | 0.784 | **0.783** | **1.113** | **1.227** | 1.303 | 1.331 | 1.343 |
| 200 | **0.820** | **0.801** | 0.789 | 0.785 | 0.782 | **1.113** | 1.230 | 1.301 | 1.328 | 1.345 |
| 500 | **0.820** | **0.801** | **0.790** | 0.785 | **0.783** | 1.114 | 1.230 | **1.299** | 1.329 | 1.343 |
| 1000 | **0.820** | **0.801** | 0.789 | 0.785 | 0.782 | 1.115 | 1.228 | 1.302 | 1.327 | 1.344 |

*Table 7.* Sensitivity analysis of $\tau$ on the CitiBike dataset ($k = 1000$). Hit ratio (HR) and cost ratio (CR) of RPB-OM under varying $\tau$ and log-normal noise parameter $\sigma$.

| $\tau$ | Hit Ratio | | | | | Cost Ratio $\downarrow$ | | | | |
|---|---|---|---|---|---|---|---|---|---|---|
| | $\sigma=5$ | $\sigma=10$ | $\sigma=20$ | $\sigma=30$ | $\sigma=50$ | $\sigma=5$ | $\sigma=10$ | $\sigma=20$ | $\sigma=30$ | $\sigma=50$ |
| 1 | 0.561 | 0.450 | 0.384 | 0.362 | 0.350 | 1.251 | 1.570 | 1.756 | 1.818 | 1.854 |
| 2 | 0.572 | 0.477 | 0.413 | 0.393 | 0.377 | 1.220 | 1.490 | 1.675 | 1.733 | 1.777 |
| 5 | 0.573 | 0.487 | 0.432 | 0.413 | 0.397 | 1.217 | 1.463 | 1.621 | 1.674 | 1.719 |
| 10 | 0.574 | 0.488 | 0.433 | **0.415** | 0.400 | 1.216 | 1.460 | 1.617 | **1.668** | 1.712 |
| 20 | 0.573 | 0.488 | **0.434** | **0.415** | 0.400 | 1.218 | 1.460 | **1.615** | 1.669 | 1.711 |
| 50 | 0.573 | 0.488 | 0.433 | 0.414 | 0.400 | 1.217 | 1.459 | 1.618 | 1.672 | 1.710 |
| 100 | **0.575** | **0.489** | **0.434** | **0.415** | 0.400 | **1.213** | **1.457** | **1.615** | **1.668** | 1.713 |
| 200 | 0.574 | **0.489** | 0.433 | 0.414 | 0.401 | 1.216 | 1.458 | 1.618 | 1.672 | 1.708 |
| 500 | 0.574 | 0.488 | 0.432 | 0.414 | 0.401 | 1.216 | 1.460 | 1.620 | 1.671 | 1.709 |
| 1000 | 0.574 | 0.487 | 0.432 | 0.414 | **0.402** | 1.216 | 1.463 | 1.619 | 1.672 | **1.707** |

We observe consistent trends across both datasets: both the hit ratio and cost ratio improve as $\tau$ increases up to around 5–10, after which the performance largely saturates. For example, on BrightKite at $\sigma = 5$, the cost ratio drops from 1.118 ($\tau = 1$) to 1.113 ($\tau = 5$); on CitiBike, it drops from 1.251 ($\tau = 1$) to 1.217 ($\tau = 5$), with negligible further improvement beyond that in both cases. This suggests that even a small value of $\tau$ (e.g., 2–10) is sufficient to achieve near-optimal performance, and that RPB-OM is not sensitive to the precise choice of $\tau$ in practice.

## K. Hit-Credit Variant of RPB-OM

We present RPB-OM-HC (Hit-Credit), an alternative rule for earning prediction budget within the RPB-ONOPT framework. The motivation is that each cache hit indicates that previous evictions kept useful pages in cache, so the algorithm can earn budget directly from cache hits.

Specifically, on each cache hit, RPB-OM-HC accumulates a fractional credit of $1/(U(\omega) + 1)$. When the accumulated credit reaches 1, the algorithm earns one unit of budget. Unlike the gate-pass condition $U(\omega) \leq (Y + 2)/e - 2$ used in RPB-OM, the hit-credit is not reset at $L_0$-misses; it persists across intervals and reflects the cumulative effectiveness of past evictions. Algorithm 4 describes RPB-OM-HC.

RPB-OM-HC retains the same theoretical guarantees as RPB-OM. In the robustness proof of Theorem 5.7, the inequality $\Delta cost + \Delta\Phi \leq 0$ at every lazy adversary request amortizes each unit of earned budget against OM's accumulated expected cost. Under the hit-credit rule, each cache hit accumulates $1/(U(\omega)+1)$, which is exactly the per-step lower bound on OM's expected cost established in Lemma 5.6. This earns budget against OM's expected cost in the same spirit as RPB-OM's gate-pass rule, so the same inequality continues to hold. The same argument extends to the smoothness proof in Theorem 5.8. Hence RPB-OM-HC is also 1-consistent, $(H_k + O(1))$-robust, and $O(1 + \log(\eta_1/cost(\text{OPT})))$-smooth.

---

**Algorithm 4** RPB-OM-HC (Hit-Credit variant)

---

1: $\omega :=$ the current work function
2: $C :=$ the current cache configuration
3: $B :=$ the prediction budget
4: $\tau :=$ the $O(1)$ budget value reset at an $L_0$-miss
5: $H := 0$ {accumulated hit-credit}
6: $V :=$ the eviction candidate set
7: **while** receiving a request to page $p$ **do**
8:   **if** $p \notin C$ **then**
9:     **if** $p \in L_0$ **then**
10:       Evict a page $x \in C$ following the predictions
11:       $B \leftarrow \tau$
12:     **else if** $p \in L_i, i > 0$ **then**
13:       Apply OM's rule to select eviction candidates: $V \leftarrow C \cap \bigcup_{l=1}^{z} L_l$, where $z = \min\left\{ j \in \{i,..,k\} \mid \right.$
            $\left. |C \cap \bigcup_{l=1}^{j} L_l| = j \right\}$
14:       **if** $H \geq 1$ **then**
15:         $B \leftarrow B + 1$
16:         $H \leftarrow H - 1$
17:       **if** $B > 0$ **then**
18:         Evict a page $x \in V$ following the predictions
19:         $B \leftarrow B - 1$
20:       **else**
21:         Evict a page $x \in V$ with the lowest OM priority
22:     Cache page $p$, update $C$
23:   **else**
24:     $H \leftarrow H + 1/(U(\omega) + 1)$

---

Table 8 compares RPB-OM-HC against RPB-OM across values of $\tau$ on the SPEC CPU 2006 benchmark. We abbreviate RPB-OM with $\tau = x$ as ROM-$x$, and RPB-OM-HC with $\tau = x$ as HC-$x$.

*Table 8.* Mean Cost Ratio of RPB-OM and RPB-OM-HC across 13 SPEC CPU 2006 datasets.

| | PLECO | | | | POPU | | | |
|---|---|---|---|---|---|---|---|---|
| | Hit Ratio (%) | | Cost Ratio ↓ | | Hit Ratio (%) | | Cost Ratio ↓ | |
| $\tau$ | ROM | HC | ROM | HC | ROM | HC | ROM | HC |
| 1 | 22.75 | 22.69 | 1.249 | 1.252 | 24.34 | 24.54 | 1.220 | 1.214 |
| 2 | 22.70 | 22.67 | 1.252 | 1.253 | 24.49 | 24.55 | 1.215 | 1.213 |
| 4 | 22.64 | 22.63 | 1.254 | 1.254 | 24.55 | 24.56 | 1.213 | 1.212 |
| 8 | 22.62 | 22.62 | 1.254 | 1.254 | 24.55 | 24.55 | 1.212 | 1.212 |
| 16 | 22.62 | 22.62 | 1.254 | 1.254 | 24.55 | 24.55 | 1.212 | 1.212 |

With the POPU predictor, RPB-OM-HC outperforms RPB-OM at small $\tau$: at $\tau = 1$, HC achieves a mean cost ratio of 1.214, comparable to ROM at $\tau = 2$. This indicates that the hit-credit rule earns roughly one additional unit of budget on average from accumulated cache hits. As $\tau$ grows, the $L_0$-reset budget alone suffices and HC's advantage diminishes.

With the PLECO predictor, the hit-credit rule does not improve over RPB-OM, consistent with PLECO's weaker prediction quality, since additional budget does not help when predictions are inaccurate.

