# OpenReview forum: "Towards Optimal Robustness in Learning-Augmented Paging"
_ICML.cc/2026/Conference — ICML 2026 spotlight_

### Official Review · Reviewer_QUL9 · 2026-03-08

**Soundness:** 3
**Presentation:** 3
**Significance:** 4
**Originality:** 3
**Overall Recommendation:** 5
**Confidence:** 2

**Summary:**

This paper considers the online paging problem with learning-augmented advice. Specifically, each time a request arrives, the algorithm is also provided with a (possibly incorrect) prediction of the next time it will arrive again. This setting has been well-studied in the learning-augmented community. The paper provides a new algorithm that achieves 1-consistency (i.e., it is 1-competitive with perfect predictions), $(H_k+O(1))$-robustness (i.e., it is never worse than $(H_k+O(1))$-competitive) and $O(\log(\frac{\eta_1}{OPT}))$-smooth which shows that the competitive ratio degrades smoothly with the $\ell_1$-error of the predictions. Consistency and robustness are optimal, while smoothness matches a previous bound of (Rohatgi, 2020).

The paper also provides experiments showing that the new algorithm is often better than previous ones.

**Compliance With Llm Reviewing Policy:**

Affirmed.

**Final Justification:**

This paper provides significant improvement over a fundamental problem in online algorithms with advice. The rebuttal addressed my comments, providing experiments also with state-of-the-art predictors. I believe this is a good submission.

**Key Questions For Authors:**

1. What are the assumptions on $\tau$? If I understand correctly, the final robustness is $H_k +\tau$. Is it possible to choose $\tau=0$ as in the experiments?
2. Does the smoothness guarantee (Theorem 5.6) hold for all choices of $\tau$?

**Limitations:**

yes

**Strengths And Weaknesses:**

**Strenghts**

S1. The idea of building on optimal online algorithms for paging, instead of Marker as previous work, is quite interesting

S2. Experiments are promising as the new algorithm performs better than previous ones in most cases.

S3. The idea of relative prediction budget might be interesting for other problems as well.

S4. The new algorithm has optimal robustness and consistency, and good smoothness, making it theoretically interesting.

**Weaknesses**

W1. The smoothness guarantees do not match the best-known guarantees of $O(\frac{\eta_1}{k \cdot OPT})$ of (Wei, 2020).

W2. In the experiments, It would have been natural to use the state-of-the-art predictor used in previous experiments of (Chledowski et al., 2021)

Typo: In Equation (3), the last equality should probably be $\geq 1$.

In summary, this paper provides new insights and ideas for online paging, which is a fundamental problem both in theory and in practice, and I think would be interesting for the learning-augmented community. My low confidence score is because I did not check all the proofs carefully.

---

> ### Author Rebuttal · Authors · 2026-03-31
>
> Thank you for your review and comments. We will add additional results and clarify the impact of $\tau$ in detail in the revision.
>
> **Weakness 1:**
>
> Since our main goal is to design learning-augmented paging algorithms built on optimal online algorithms while simultaneously achieving optimal robustness and consistency, in our framework smoothness is obtained as an additional guarantee, though not at the best-known level. We also note that Wei’s $O(\eta_1/(k \cdot OPT))$-smooth BlindOracle\&LRU loses 1-consistency, while the $O((\log k)/k \cdot \eta_1/OPT)$-smooth LNonMarker loses bounded robustness. Therefore, further improving smoothness while maintaining the current optimal consistency and robustness guarantees, and exploring whether this is possible or at what cost, can be an interesting direction for future work.
>
> **Weakness 2:**
>
> Thank you for this suggestion. We implemented and trained the PARROT-CACHE predictor used by Chledowski et al. [2] on several datasets. This predictor estimates which page would be evicted from the cache under Belady’s rule. We evaluate the algorithms using this predictor by following its suggestion whenever the algorithm invokes it. Specifically, RPB-OM follows the predictor whenever it uses predictions (Line 16 of Algorithm 3). The results are as follows. For completeness, we will add more results, as well as results for another predictor, PARROT-REUSE, in the revision.
>
> | Dataset | LRU | BlindOracle | LMarker | PredictiveMarker | BlindOracle\&LRU | RPB-OM($\tau=0$) | RPB-OM($\tau=1$) | RPB-OM($\tau=2$) |
> |---|---|---|---|---|---|---|---|---|
> | xalanc | 45.08\% | 39.10\% | 44.36\%  | 44.71\% | 44.31\% | 41.75\% | 43.51\% | 44.02\% |
> | bzip | 63.81\% | 64.93\% | 64.41\%  | 64.90\% | 64.92\% | 63.75\% | 64.79\% | 65.16\% |
> | bwaves | 0.00\% | 0.06\% | 0.00\%  | 0.00\% | 0.06\% | 0.09\% | 0.1\% | 0.1\% |
> | cactusadm | 0.00\% | 14.82\% | 1.76\%  | 1.77\% | 14.82\% | 8.83\% | 10.59\% | 11.62\% |
> | leslie3d | 9.49\% | 7.51\% | 10.10\%  | 9.66\% | 9.22\% | 18.43\% | 18.09\% | 18.03\% |
> | milc | 0.01\% | 0.71\% | 0.01\%  | 0.01\% | 0.69\% | 0.3\% | 0.24\% | 0.18\% |
>
>
> Also, thank you for catching the typo in Equation (3). It will be fixed in the revision.
>
> **Question 1:**
>
> The framework performs a prediction-driven eviction by default on each $L_0$-miss, and then resets $B$ to $\tau$, which serves as the budget for subsequent prediction-driven evictions within that phase. Thus, the default prediction-driven eviction remains effective even when $\tau = 0$. Accordingly, $\tau$ should be interpreted as the *additional* prediction budget available within each phase after the initial $L_0$-miss.
>
> Under this assumption, the robustness guarantee is in fact $H_k + \tau + 1$. Correspondingly, Equation (4) near Line 363 should be revised to $\mathbb{E}\Big[\sum_{i=1}^{m} \Delta_{t_i}\Big] \le \tau + 1$, and we will correct this in the revision.
>
> Therefore, choosing $\tau = 0$ is valid, and this is exactly the setting used in our experiments. In that case, the robustness bound becomes $H_k + 1$, which matches the lower bound for $1$-consistent learning-augmented paging algorithms. If one relaxes $1$-consistency, the robustness bound can be made closer to $H_k$, but it still cannot equal $H_k$ as long as the algorithm relies on predictions. Achieving exactly $H_k$ would require ignoring predictions altogether, in which case the algorithm would no longer be meaningfully learning-augmented.
>
> **Question 2:**
>
> The smoothness bound holds as long as $\tau = O(1)$, i.e., $\tau$ is a fixed constant. In particular, it holds for all choices of $\tau$ used in our experiments, including $\tau=0$.  A large $\tau$ invalidates the current smoothness bound. Intuitively, since the offline optimum (OPT) incurs only one miss each phase started by an $L_0$-miss, $\tau$ must also be restricted to $O(1)$. This differs from Marker-based learning-augmented algorithms such as PredictiveMarker. Under marker-based phases, OPT incurs $O(x)$ cost within a phase, where $x$ is the number of clean pages requested in that phase. We will add a discussion in the appendix comparing our algorithm with existing ones to make the role of $\tau$ more explicit.
>
> **References**
>
> [1] Wei, A. Better and Simpler Learning-Augmented Online
> Caching. In Byrka, J. and Meka, R. (eds.), Approximation, Randomization, and Combinatorial Optimization. Algorithms and Techniques (APPROX/RANDOM 2020), volume 176 of Leibniz International Proceedings in Informatics (LIPIcs), pp. 60:1–60:17, Dagstuhl, Germany, 2020.
>
> [2] Jakub Chłe ̨dowski, Adam Polak, Bartosz Szabucki, and Konrad Tomasz Zołna. Robust learningaugmented caching: An experimental study. In Marina Meila and Tong Zhang, editors, Proceedings of the 38th International Conference on Machine Learning, volume 139 of Proceedings of Machine Learning Research, pages 1920–1930. PMLR, 18–24 Jul 2021.

---

> > ### Author Rebuttal · Reviewer_QUL9 · 2026-03-31
> >
> > I thank the authors for their response. My concerns have been addressed, so I maintain my already positive score.

---

> > > ### Author Response · Authors · 2026-04-01
> > >
> > > Thank you for your reply and acknowledgement!

---

### Official Review · Reviewer_GGeD · 2026-03-10

**Soundness:** 3
**Presentation:** 3
**Significance:** 3
**Originality:** 3
**Overall Recommendation:** 5
**Confidence:** 3

**Summary:**

This paper investigates the robustness and consistency issues of  learning-augmented paging algorithms , proposes a novel algorithmic framework called  RPB-ONOPT , and implements an algorithm  RPB-ONLINE MIN  based on this framework. This algorithm achieves  $H_k + O(1)$-robustness  while maintaining  1-consistency , approaching the theoretical optimum.

**Compliance With Llm Reviewing Policy:**

Affirmed.

**Final Justification:**

The paper makes a meaningful and technically solid contribution to learning-augmented paging by introducing a novel unifying perspective and achieving strong robustness guarantees. I believe the core results are sound and that this work is worthy of acceptance.

**Key Questions For Authors:**

1. Although RPB is an intuitive ``budget'' concept, the paper does not provide its mathematical definition or formal model. For example: Does the budget update rule satisfy some form of invariance? How do the upper and lower bounds of the budget affect robustness?

2. In particular, the proof of Theorem~5.5 involves multiple probabilistic analyses and inequality derivations, but some steps lack intermediate explanations.

3. While comparisons are made with LMARKER, FR, etc., there is no direct comparison with enhanced versions of  OPT  or  Belady's algorithm  (such as Belady's algorithm with predictions).

4. The choice of $\tau \in \{0,1,2\ $ significantly impacts performance, but the rationale behind selecting these values is not explained, nor is a sensitivity analysis provided.

5. Some sentences are overly complex, affecting readability. For example, the description of RPB-OM in Section~5.2.

**Limitations:**

yes

**Strengths And Weaknesses:**

1. Introduces the concept of  RPB , unifying the design ideas of various learning-augmented paging algorithms.
2. Proves that RPB-ONLINE MIN exhibits both  1-consistency  and  $H_k + O(1)$-robustness , which is currently the best-known robustness result.
3. Provides a new proof for ONLINE MIN, facilitating subsequent algorithmic analysis.

---

> ### Author Rebuttal · Authors · 2026-03-31
>
> Thank you for your review and valuable questions. We welcome any further discussion or suggestions.
>
> **Question 1:**
>
> We agree that a clearer formulation would improve the presentation. Below is a draft that we plan to add in the revision.
>
> Formally, given a robust baseline algorithm $A$, the RPB mechanism maintains a nonnegative budget $B_t$ that controls whether the algorithm may deviate from $A$. At time $t$, $B_{t+1}=B_t+\textit{earn}_t-\textit{spend}_t$, where $\textit{earn}_t,\textit{spend}_t\in \{0,1\}$. Here, \(\textit{earn}_t = 1\) when recent eviction decisions perform at least as well as the baseline, and \(\textit{spend}_t = 1\) when \(B_t > 0\) and the algorithm takes a prediction-driven eviction. The budget is reset to $\tau$ at each $L_0$-miss.
>
> Thus, the invariant $0 \le B_t \le \tau$ always holds. The lower bound ensures that prediction-driven evictions occur only when justified by past prediction effectiveness, thereby preventing unbounded cost from prediction errors, while the upper bound limits the extra cost over OM to at most $\tau=O(1)$ per phase. Both contribute to the $H_k + O(1)$ robustness bound.
>
> **Question 2:**
>
> We will rewrite the proof from Line 377 to Equation (3) to improve clarity:
>
> Let $t_e$ be a miss time in the phase, and let $V_{t_e}$ be OM’s eviction-candidate set at time $t_e$, with $s := |V_{t_e}|$. By definition, OM assigns the pages in $V_{t_e}$ a uniformly random priority order and evicts the minimum-priority page.
>
> Now consider the later re-requests to pages in $V_{t_e}$. Suppose the $i$-th such page is requested at time $t_i > t_e$. By then, exactly $i-1$ pages in $V_{t_e}$ have already been revealed, so $s-i+1$ remain unrevealed. OM misses at $t_i$ exactly when the requested page is the minimum-priority page among these $s-i+1$ pages. Since the initial priority order is uniform, each remaining page is equally likely to be the minimum, and hence $\Pr[\text{OM misses at } t_i] = 1/(s-i+1)$.
>
> By Lemma 5.4, each lazy request to an unrevealed page decreases the candidate-set size by exactly $1$. Therefore, after $m = s-\frac{s}{e}+1$ such requests, the candidate-set size becomes at most $s/e-1$. The expected number of OM misses on these first $m$ requests is
>
> \begin{equation*}
> \sum_{i=1}^{m}\Pr[\text{OM misses at } t_i]
> = \sum_{i=1}^{m}\frac{1}{s-i+1}
> = H_s-H_{s-m}
> = \sum_{j=s-m+1}^{s}\frac{1}{j}.
> \end{equation*}
>
> Since $\int_j^{j+1}\frac{dx}{x}\le \frac{1}{j}$ for every integer $j$, we have
>
> $$
> \sum_{i=1}^{m}\Pr[\mathrm{OM\ misses\ at}\ t_i]
> \ge \int_{s-m+1}^{s+1}\frac{dx}{x}
> = \ln\frac{s+1}{s-m+1}
> = \ln\frac{s+1}{s/e}
> = \ln\left(e\cdot\frac{s+1}{s}\right)
> \ge 1.
> $$
>
> **Question 3:**
>
> There may be a misunderstanding here. In paging, Belady’s algorithm (OPT) is already the offline optimum and has full knowledge of the future request sequence. Thus, “Belady with predictions” is not a meaningful separate benchmark. Accordingly, we use OPT/Belady as the offline baseline and compare against prior online and learning-augmented paging algorithms.
>
> **Question 4:**
>
> Thank you for this suggestion. From a theoretical perspective, $\tau$ must remain $O(1)$ to preserve the $H_k + O(\tau) = H_k + O(1)$ robustness guarantee, as well as the current smoothness bound. In our experiments, we set $\tau \in \{0,1,2\}$ because the SPEC CPU2005 benchmark requires a relatively small cache size, namely $k=16$. The table below analyzes the sensitivity to $\tau$ over a broader range of values.
>
> | Metric | LRU | BlindOracle | RPB-OM($\tau=0$) | RPB-OM($\tau=1$) | RPB-OM($\tau=2$) | RPB-OM($\tau=4$) | RPB-OM($\tau=8$) | RPB-OM($\tau=16$) |
> |---|---|---|---|---|---|---|---|---|
> | Mean Hit (\%) | 14.27 | 15.92 | 22.46 | 22.72 | 22.70 | 22.63 | 22.62 | 22.62 |
> | Std Hit | 19.38 | 20.41 | 22.39 | 22.32 | 22.27 | 22.18 | 22.18 | 22.18 |
>
> Because $k$ is small, above results with different $\tau$ are similar. We therefore evaluate cost ratios on the commonly used Brightkite dataset with $k = 1000$ to better test the sensitivity to $\tau$. Log-normal noise is added to the true next-request times to generate synthetic predictions. The results below show that a small $\tau$ relative to $k$ is preferable, while a large $\tau$ can hurt robustness.
>
> |Algorithm|LogNoise-0|LogNoise-5|LogNoise-10|LogNoise-20|LogNoise-30|
> |---|---|---|---|---|---|
> |LRU|1.178|1.178|1.178|1.178|1.178|
> |BlindOracle|1.000|1.108|1.190|1.235|1.259|
> |RPB-OM($\tau=0$)|1.000|1.159|1.230|1.242|1.255|
> |RPB-OM($\tau=1$)|1.000|1.122|1.191|1.219|1.227|
> |RPB-OM($\tau=10$)|1.000|1.120|1.181|1.209|1.226|
> |RPB-OM($\tau=20$)|1.000|1.120|1.174|1.205|1.212|
> |RPB-OM($\tau=50$)|1.000|1.112|1.179|1.201|1.207|
> |RPB-OM($\tau=100$)|1.000|1.113|1.181|1.219|1.207|
> |RPB-OM($\tau=500$)|1.000|1.110|1.193|1.215|1.212|
> |RPB-OM($\tau=1000$)|1.000|1.121|1.192|1.217|1.218|
>
>
> **Question 5:**
>
> Thank you and we will simplify the sentences near Line 320 describing RPB-OM to improve clarity.

---

> > ### Author Rebuttal · Reviewer_GGeD · 2026-04-01
> >
> > I thank the authors for their response. The rebuttal resolves enough of my earlier concerns that I am updating my recommendation to Weak Accept.

---

> > > ### Author Response · Authors · 2026-04-01
> > >
> > > Thank you for your reply! We are glad to hear our response was helpful.

---

### Official Review · Reviewer_8By5 · 2026-03-12

**Soundness:** 3
**Presentation:** 4
**Significance:** 4
**Originality:** 3
**Overall Recommendation:** 5
**Confidence:** 4

**Summary:**

Paging is one of the most fundamental problems in the field of competitive analysis, and it is also one of the first and most studied problems in the learning-augmented framework. Without predictions, the optimal competitive ratio is known to be $H_k$, where $k$ is the cache size and $H_k$ the $k$-th harmonic number. In the setting with predictions, it has been established since the first papers studying this problem that we can achieve $O(H_k)$-robustness and $O(1)$-consistency, which is an optimal tradeoff up-to multiplicative constants. The current paper explores what are the best constants that we can obtain, and it proves an algorithm that is $(H_k + O(1))$-robust and $1$-consistent. This constitutes an important improvement over the best known $2H_k$-robust and $1$-consistent algorithm.

**Compliance With Llm Reviewing Policy:**

Affirmed.

**Final Justification:**

The contribution of the paper is strong and interesting, and the authors addressed my questions, so I maintain my recommendation for acceptance.

**Key Questions For Authors:**

1. Can the approach and ideas of the paper be applied to variants of the learning-augmented paging? For example in the setting where the predictions are parsimonious?
2. Similarly, can be generalized to other related problems? (for e.g. some other problems on metrical task systems)
3. Can the authors provide intuition on the optimality or sub-optimality of the smoothness bound? Are there possible future directions to improve the algorithms proposed in the paper? (maybe on aspects beyond robustness/smoothness such as smoothness/computational efficiency,...)

**Limitations:**

yes

**Strengths And Weaknesses:**

### Strengths
1. The paper has many strong technical contributions:
   - it provides a new proof of the competitive ratio $H_k$ for OnlineMin (in the setting without predictions)
   - provides a unifying framework in the learning-augmented setting to understand how bounded robustness can be achieved under 1-consistency
   - Improves upon state-of-the art bounds in learning-augmented paging, and provides an algorithm that is Pareto-optimal up to additive constant in robustness, which is a very strong result, especially considering that extensive research has already been conducted on paging with predictions
2. The paper is well structured and easy to follow
3. The authors validate their theoretical findings by showing that their algorithms improve upon prior ones also in simulations, showing that their findings are also of practical interest

### Weaknesses
I do not see any major weaknesses in the paper, Below are some minor ones
1. How to choose the parameter $\tau$ in practice is not well explored. The paper provides experimental results with $\tau \in$ {0,1,2}, but I think this is not enough to study sensitivity of the algorithm to this parameter. More extensive experiments, testing also with large values of $\tau$ (violating the assumption$ could be interesting to better understand its impact
2. Figure 2 and table 1 only report average values over the experiments, but do not report uncertainty measures (std or other). These are essential and I encourage the authors to include them in the results.

---

> ### Author Rebuttal · Authors · 2026-03-31
>
> Thank you for your review and valuable feedback. We will include additional results in the revision.
>
> **Weaknesses 1 and 2:**
>
> We originally set $\tau$ for theoretical reasons: as long as $\tau = O(1)$, the robustness bound remains unchanged. However, we agree that $\tau$ matters in practice, so we tested larger values on the SPEC CPU2006 benchmark using the PLECO predictor. The table below also includes standard deviations.
>
> | Algorithm | Mean Hit (\%) | Std Hit | Mean CR | Std CR |
> |-----------|----------|---------|---------|--------|
> | LRU | 14.27 | 19.38 | 1.478 | 0.615 |
> | OnlinMin | 16.33 | 19.39 | 1.395 | 0.316 |
> | BlindOracle | 15.92 | 20.41 | 1.404 | 0.390 |
> | F\&R | 17.87 | 20.45 | 1.360 | 0.298 |
> | LMarker | 18.13 | 23.30 | 1.335 | 0.250 |
> | PredictiveMarker | 18.10 | 22.29 | 1.335 | 0.251 |
> | OnOPT-OM | 22.42 | 22.43 | 1.254 | 0.251 |
> | RPB-OM($\tau=0$) | 22.46 | 22.39 | 1.253 | 0.251 |
> | RPB-OM($\tau=1$) | 22.72 | 22.32 | 1.251 | 0.260 |
> | RPB-OM($\tau=2$) | 22.70 | 22.27 | 1.252 | 0.264 |
> | RPB-OM($\tau=4$) | 22.63 | 22.18 | 1.254 | 0.269 |
> | RPB-OM($\tau=8$) | 22.62 | 22.18 | 1.255 | 0.269 |
> | RPB-OM($\tau=16$) | 22.62 | 22.18 | 1.255 | 0.269 |
> | OPT  | 34.62 |  24.60 |   1.000 |   0.000 |
>
> For the SPEC CPU2006 benchmark, $k$ is required to be set to $16$, which is relatively small. To better test the sensitivity to $\tau$, we therefore also evaluate cost ratios on the commonly used Brightkite dataset with $k = 1000$, and log-normal noise is added to the true next-request times to generate synthetic predictions. The results below show that a small $\tau$ relative to $k$ is preferable, while a large $\tau$ can hurt robustness.
>
> | Algorithm | LogNoise-0 | LogNoise-5 | LogNoise-10 | LogNoise-20 | LogNoise-30 |
> |---|---|---|---|---|---|
> | LRU | 1.178 | 1.178 | 1.178 | 1.178 | 1.178 |
> | BlindOracle | 1.000 | 1.108 | 1.190 | 1.235 | 1.259 |
> | RPB-OM($\tau=0$) | 1.000 | 1.159 | 1.230 | 1.242 | 1.255 |
> | RPB-OM($\tau=1$) | 1.000 | 1.122 | 1.191 | 1.219 | 1.227 |
> | RPB-OM($\tau=10$) | 1.000 | 1.120 | 1.181 | 1.209 | 1.226 |
> | RPB-OM($\tau=20$) | 1.000 | 1.120 | 1.174 | 1.205 | 1.212 |
> | RPB-OM($\tau=50$) | 1.000 | 1.112 | 1.179 | 1.201 | 1.207 |
> | RPB-OM($\tau=100$) | 1.000 | 1.113 | 1.181 | 1.219 | 1.207 |
> | RPB-OM($\tau=500$) | 1.000 | 1.110 | 1.193 | 1.215 | 1.212 |
> | RPB-OM($\tau=1000$) | 1.000 | 1.121 | 1.192 | 1.217 | 1.218 |
>
>
> **Question 1:**
>
> Yes. Prior work has considered parsimonious prediction by reducing either the number of prediction candidates or the number of predictor calls. Our approach uses predictions only on each $L_0$ miss, the minimum required for 1-consistency (see Appendix C), while the relative prediction budget further gates prediction-driven actions to avoid unnecessary predictor calls. These ideas may help refine other learning-augmented paging algorithms that aim to limit predictor usage. In addition, the structural properties of the work function suggest that restricting the eviction candidate set between two consecutive $L_0$ requests may be worth exploring, since predictions must be erroneous in this case.
>
> **Question 2:**
>
> The design principle of budgeting prediction‑driven deviations from a robust baseline is applicable beyond paging. The relative prediction budget can serve as a mechanism for controlling such deviations. However, general MTS allows arbitrary task cost vectors over a metric space, making the decision space significantly richer than in paging.
>
> In our paper, the key ingredient is a metric for comparing eviction decisions to determine how the prediction budget is charged. If a similar metric can be developed for MTS, the same high-level idea may still apply. Exploring this is an interesting direction for future work.
>
> **Question 3:**
>
> Our smoothness bound is not known to be optimal. Rohatgi’s LNonMarker (SODA 2020) achieves the stronger bound $O\left(1+\log(k)/ k \cdot \eta_1 / OPT\right)$, which can be asymptotically better when the relative prediction error is small. For example, if $\eta_1/OPT=k / \log k$, our bound is $O(\log k)$, while LNonMarker achieves $O(1)$. However, LNonMarker does not by itself provide a robustness guarantee and must be combined with a classical algorithm to recover worst-case competitiveness. In contrast, our algorithm directly maintains optimal robustness $H_k + O(1)$ while achieving logarithmic smoothness.
>
> Rohatgi also proves a lower bound of $\Omega\left(\log \min\left(1/(k\log k) \cdot \eta_1/OPT, k\right)\right)$ on the competitive ratio as a function of prediction error. But no algorithm currently matches this lower bound in all regimes. Bridging this gap remains an interesting open problem.
>
> Also, LNonMarker improves smoothness at the cost of bounded robustness. It remains unclear whether better smoothness is possible without such trade-offs, or whether the lower bound is unattainable. Other directions include more parsimonious use of predictions and extensions to more general caching models.

---

> > ### Author Rebuttal · Reviewer_8By5 · 2026-04-01
> >
> > Thank you for addressing my questions and for the additional experiments!
> > I maintain my positive score.

---

> > > ### Author Response · Authors · 2026-04-02
> > >
> > > We appreciate your reply and are glad to know that our response was helpful.

---

### Official Review · Reviewer_8r3F · 2026-03-22

**Soundness:** 3
**Presentation:** 4
**Significance:** 4
**Originality:** 4
**Overall Recommendation:** 5
**Confidence:** 4

**Summary:**

This paper studies learning-augmented paging and closes the robustness gap for randomized caching algorithms. Its main contribution is RPB-ONOPT, a framework based on a new notion called relative prediction budget, which controls when the algorithm should trust predictions versus fall back to a robust baseline. Using this framework, the authors design RPB-OM, which achieves 1-consistency under perfect predictions and optimal robustness up to constants, namely $H_k + O(1)$, improving on prior bounds like $2H_k + O(1)$. They also provide a new proof of ONLINEMIN’s $H_k$-competitiveness and experiments on SPEC CPU2006 traces showing strong empirical performance.

**Compliance With Llm Reviewing Policy:**

Affirmed.

**Key Questions For Authors:**

1. How does it work on more realistic algorithms such as LRB?
2. The prediction error assumes that it is equal likely to make mistakes in small and large reuse distance for different objects, however, predicting difficulty varies for different objects. An object that gets reused every 10 requests is easier to predict than an object that gets a reuse every 1000 requests.
3. Would the idea applicable to LeCaR and other expert based algorithms?

**Strengths And Weaknesses:**

Strengths and Weaknesses
+ clear writing with sufficient background
+ solving an important problem
+ strong theoretical contribution
- limitation on evaluation

---

> ### Author Rebuttal · Authors · 2026-03-31
>
> Thank you for your review and your valuable questions. The additional experimental results below will be included in the revised paper, and we would be happy to discuss any further questions.
>
> **Question 1:**
>
> The eviction priorities in LRB are binary [1]. Following the corresponding paper, we implemented (i) the original LRB, (ii) LRB\&LRU, which replaces BlindOracle in the BlindOracle&LRU algorithm, and (iii) RPB-OM-LRB, which incorporates LRB’s eviction policy into RPB-OM when using predictions; that is, Line 16 in Algorithm 3 is replaced with LRB’s eviction rule. The experimental results are as follows:
>
> | Metric | OPT | LRU | Marker | OnlineMin | LRB | LRB\&LRU | RPB-OM-LRB ($\tau=0$) | RPB-OM-LRB ($\tau=1$) | RPB-OM-LRB ($\tau=2$) |
> |---|---|---|---|---|---|---|---|---|---|
> | Avg Cost Ratio | 1.000 | 1.478 | 1.394 | 1.398 | 1.372 | 1.339 | 1.271 | 1.267 | 1.264 |
> | Avg Hit Rate | 34.6\% | 14.3\% | 16.4\% | 16.3\% | 17.2\% | 18.37\% | 23.9\% | 24.0\% | 24.1\% |
>
>
> **Question 2:**
>
> We agree that the difficulty of predicting different objects can vary. In this paper, we use the $\ell_1$ error of the predicted next-request times as the prediction error metric, as it is widely adopted in related work [2, 3]. Prior work has also studied the $\ell_2$ error of predicted next-request times and classification error for binary predictions, but these metrics are defined independently of the objects themselves.
>
> Your comment inspired us to consider whether prediction difficulty should be incorporated into the error measure. Doing so may require accounting for learnability and could broaden algorithm design to jointly consider prediction and the learnability of specific objects. Moreover, since current metrics such as consistency and robustness follow the traditional competitive-ratio framework, incorporating learnability or prediction difficulty may also inspire new metrics that are more closely connected to machine learning. We believe this is an interesting direction for future research.
>
> **Question 3:**
>
> Yes. According to the paper[2], LeCaR is an online learning-based caching algorithm that maintains two experts, LRU and LFU, and dynamically adjusts trust in each based on their historical performance. LeCaR performs online learning--style algorithm weight updates by checking the optimality of eviction decisions. Its weight update rule penalizes an algorithm whenever it is selected by LeCaR and makes a suboptimal eviction decision, which can lead to unnecessary switching even when that suboptimal algorithm consistently performs better than another.
>
> Therefore, the idea of a relative prediction budget can be applied to LeCaR. By introducing a relative prediction budget, the weight update rule can be refined by additionally considering the relative effectiveness of different eviction decisions. For example, suppose the currently missed item $x$ was evicted under algorithm candidate A at time $t$, while another item $y$ would have been evicted had algorithm candidate B been followed at time $t$. By checking whether item $y$ has been requested during the interval from time $t$ to the present, we can infer the relative eviction effectiveness of algorithms A and B at time $t$, namely, the relative ordering of the next-request times of $x$ and $y$ at time $t$. Based on this, we can choose to charge the budget of an algorithm candidate if it performs better than the alternative, and penalize it otherwise.
>
> The above provides an example based on LeCaR. The idea of a fine-grained comparison of eviction effectiveness (as discussed in Section 4) underlying the relative prediction budget may also be applied to other existing algorithms and could be explored in future work.
>
> **References**
>
> [1] Song, Zhenyu, et al. "Learning relaxed belady for content distribution network caching." 17th USENIX Symposium on Networked Systems Design and Implementation (NSDI 20). 2020.
>
> [2] Lykouris, Thodoris, and Sergei Vassilvitskii. "Competitive caching with machine learned advice." Journal of the ACM (JACM) 68.4 (2021): 1-25.
>
> [3] Rohatgi, Dhruv. "Near-optimal bounds for online caching with machine learned advice." Proceedings of the Fourteenth Annual ACM-SIAM Symposium on Discrete Algorithms. Society for Industrial and Applied Mathematics, 2020.
>
> [4] Vietri, Giuseppe, et al. "Driving cache replacement with {ML-based}{LeCaR}." 10th USENIX Workshop on Hot Topics in Storage and File Systems (HotStorage 18). 2018.

---

> > ### Author Rebuttal · Reviewer_8r3F · 2026-03-31
> >
> > Thank you for the response!
> > A minor comment: I believe you should use hit ratio or miss ratio instead of rate (which has a unit).

---

> > > ### Author Response · Authors · 2026-04-01
> > >
> > > Thank you for your response and for the additional suggestion!

---

### Decision · Program_Chairs · 2026-04-30

**Decision:**

Accept (spotlight)

**Comment:**

The paper studies learning-augmented paging, where a cache replacement algorithm can use machine-learned predictions while still maintaining worst-case guarantees. It focuses on closing the gap between the best known robustness bounds for randomized learning-augmented paging and the optimal competitive ratio $H_k$ from classical online paging. The authors introduce a unifying concept, the relative prediction budget, to explain how prior robust algorithms decide when it is safe to trust predictions. Their analysis suggests that previous methods either overuse predictions, which hurts robustness, or underuse them, which hurts practical performance. Based on this, they propose a new framework, RPB-ONOPT, and instantiate it as RPB-OM by combining prediction guidance with ONLINEMIN-style online-optimal structure. The main theoretical result is that RPB-OM achieves $1$-consistency and $H_k + O(1)$-robustness, which is the best possible robustness up to an additive constant. The paper also proves a smoothness guarantee matching the best known logarithmic dependence on prediction error. Finally, experiments on SPEC CPU 2006 cache traces with both synthetic and real predictors show that RPB-OM performs strongly in practice and generally improves over prior learning-augmented paging methods.

All reviewers agree that the paper makes strong contributions and did not identify any major weaknesses. The only concerns raised were that the smoothness guarantees do not match the current state of the art, and that the paper would benefit from additional numerical evaluation. The authors provided convincing responses to these points and supplemented the paper with further experimental results.